# Learning to Rank by Directly Optimizing Full-Order Probabilities

**Yongxiang Tang** [1]   **Chao Wang** [1]   **Jincheng Lu** [1]   **Yanhua Cheng** [1]   **Xialong Liu** [1]   **Peng Jiang** [1]

## Abstract

Learning to rank can be cast as a probabilistic modeling problem over permutations, where the goal is to estimate the likelihood of an observed total ordering of items. This formulation naturally involves full-order probabilities of the form $\mathbb{P}(z_1 \leq \cdots \leq z_n)$, whose exact computation and optimization are intractable due to the factorial growth of the permutation space with respect to the list size. In this work, we introduce the *Full-Order Bound* (FOB), a tractable lower bound on the probability of an observed ordering, constructed from a subset of ordering constraints that factorizes across items while preserving full-order structure and order-reversal invariance. Under log-concave latent densities, the bound induces a convex inner tightening problem over latent cut points, which we solve efficiently during training using a *safe-region gradient ascent* (SRGA) procedure. Experiments on synthetic ranking tasks and large-scale learning-to-rank benchmarks show that FOB improves full-list ordering metrics and remains competitive on NDCG, while an optional metric-aligned variant recovers NDCG gains. Our code is available at https://github.com/tyxaaron/FOB.

## 1. Introduction

Learning to rank is a central problem in information retrieval, recommendation, and advertising systems (Li, 2011). Given a set of items with relevance signals, the goal is to learn a scoring function $f_\theta$ whose induced ordering aligns with observed preferences. From an order-level probabilistic perspective, each training instance corresponds to an ordering event, and an ideal model would assign high full-order probability to the observed ordering. However, the number of ordering events grows factorially with the list size, so

directly modeling and optimizing full-order probabilities is intractable for realistic problems.

Facing this difficulty, most practical learning-to-rank objectives fall into two broad categories. The first category consists of *surrogate* objectives defined on lower-order statistics. Pairwise and listwise surrogates (Burges, 2010) reduce ranking to losses over score comparisons or position-wise gains, while differentiable sorting methods (Grover et al., 2019; Petersen et al., 2022) replace the discrete permutation operator with a continuous map and optimize losses defined on relaxed outputs. These approaches are computationally efficient and empirically effective, but they deliberately abandon full-order probabilities. We show that this compression induces a *gradient-level non-separability*: for any low-dimensional ($\dim < n! - 1$) surrogate representation, there exist distinct permutation distributions that induce identical expected gradients for all score vectors, so the learning dynamics cannot distinguish between them (Section 3.2 and Appendix A).

The second category consists of *probabilistic* models that attempt to represent full-order probabilities explicitly. Representative methods such as ListMLE and ListNet (Xia et al., 2008; Cao et al., 2007) assign probabilities to permutations via Plackett–Luce (PL) style sequential factorizations, where items are selected one at a time according to their scores. This strategy retains a full-order likelihood but introduces an implicit directionality tied to a particular generation order. In contrast, ordering events admit a natural order-reversal symmetry: reversing the labels and negating the scores should leave the learning problem unchanged. We formalize this requirement as an *order-reversal invariance* condition, defined in Section 3.3, and show that standard PL-type objectives violate it in general.

In this work, we propose the *Full-Order Bound* (FOB), which directly targets full-order probabilities of the form $\mathbb{P}(z_1 \leq \cdots \leq z_n)$ while addressing both limitations discussed above. We start from a latent score model and introduce a set of scalar cut points per instance that partition the real line into ordered intervals. Given an observed ordering, we construct a refined ordering event by constraining each latent score to lie within its corresponding interval, which yields a tractable lower bound on the full-order probability. The bound factorizes across items, preserves order-reversal

---

[1]Kuaishou Technology. Correspondence to: Yongxiang Tang <tangyongxiang94@gmail.com>.

*Proceedings of the 43rd International Conference on Machine Learning*, Seoul, South Korea. PMLR 306, 2026. Copyright 2026 by the author(s).

invariance by design, and is designed to preserve full-order structure rather than explicitly compressing supervision into pairwise or position-wise statistics (Section 4).

We show that tightening FOB leads to a bilevel structure: for fixed model parameters, we tighten the bound by adjusting the cut points, and then update the model using the tightened objective. We further develop a safe-region gradient ascent (SRGA) procedure that tightens the bound while enforcing ordering constraints on the cut points, and combine it with a simple list-length warmup scheme to stabilize early training (Section 5.3 and Appendix E). Beyond SRGA itself, the training procedure introduces very few additional hyper-parameters, and the inner updates remain computationally efficient in practice (Appendix M).

Experiments on synthetic ranking tasks and large-scale learning-to-rank benchmarks demonstrate that FOB is competitive with strong baselines and often improves full-list consistency, with especially clear gains on synthetic long-list settings. These results highlight the practical benefits of explicitly modeling full-order structure (Section 6). For completeness, we also report an optional metric-aligned variant (LAMBDAFOB) for comparison with Lambda-style objectives (Section 6.2 and Appendix G).

Our contributions are summarized as follows:

- We provide an analysis of learning-to-rank objectives and identify two structural limitations in existing methods: gradient-level non-separability of low-dimensional surrogates and order-reversal asymmetry of sequential factorization models.

- We introduce the Full-Order Bound, a tractable lower bound on full-order probabilities constructed via shared cut points that factorize across items while preserving order-reversal symmetry. We conjecture that the tightened Gaussian FOB family has full permutation rank and provide numerical evidence for list sizes up to $n = 6$.

- We show that tightening FOB leads to a convex inner problem over cut points and propose a safe-region gradient ascent procedure for efficient and stable training.

- We demonstrate consistent gains on synthetic ranking tasks and competitive large-scale benchmark results, especially on full-list consistency metrics.

## 2. Related Work

Learning to rank has been studied extensively in information retrieval and machine learning (Li, 2011; Burges, 2010). Over the years, a variety of objectives and modeling paradigms have been proposed. From the perspective of order-level probabilities, most existing approaches can

be grouped into two broad categories: *surrogate* methods that optimize losses defined on lower-order statistics, and *probabilistic* models that assign likelihoods to full-order events. We briefly review these directions and discuss their limitations.

**Surrogate objectives.** Surrogate methods optimize losses defined on pairwise or listwise statistics instead of full-order probabilities. Representative approaches include RankNet, LambdaRank, and their variants (Herbrich et al., 2000; Burges et al., 2005; 2006; Qin et al., 2008; 2010; Huang et al., 2013; Schroff et al., 2015). These methods assign each item a deterministic relevance score and optimize losses defined over pairwise or listwise surrogates. A representative formulation is RankNet, which models

$$\mathbb{I}(y_i \le y_j) \approx \mathbb{P}(y_i \le y_j) = \sigma(s_j - s_i),$$

where $s_i$ and $s_j$ are the scores assigned to items by the model, and minimizes a cross-entropy loss over labeled preference pairs. LambdaRank and related gradient-based listwise surrogates adjust such pairwise gradients by position-dependent weights to better align with metrics such as NDCG (Burges et al., 2006; Qin et al., 2010).

A complementary line of work makes the sorting operation itself differentiable in order to enable end-to-end gradient-based optimization (Grover et al., 2019; Swezey et al., 2021; Petersen et al., 2022; Kim et al., 2024). In its abstract form, sorting can be expressed as

$$\text{sort}(\mathbf{s}) = \mathbf{\Pi}(\mathbf{s})\,\mathbf{s} \approx \hat{\mathbf{\Pi}}(\mathbf{s})\,\mathbf{s},$$

where $\mathbf{\Pi}(\mathbf{s})$ is the permutation matrix induced by scores and $\hat{\mathbf{\Pi}}(\mathbf{s})$ is a continuous relaxation. Training objectives are then defined on losses between $\hat{\mathbf{\Pi}}(\mathbf{s})$ and a target permutation matrix $\mathbf{\Pi}(\mathbf{y})$, or on downstream utilities computed from the relaxed outputs. Although these approaches allow gradients to propagate through sorting-like operators and have shown strong empirical performance, they deliberately optimize surrogates defined on low-dimensional ($O(n^2)$) summaries of the ordering event. Collections of pairwise, position-wise, or relaxed permutation statistics do not uniquely determine the full-order probability of an observed ordering event among the $n!$ possibilities.

**Probabilistic models.** Probabilistic listwise methods attempt to model ordering outcomes more directly by assigning probabilities to permutations. Representative approaches such as ListNet, ListMLE, and BayesRank (Cao et al., 2007; Xia et al., 2008; Kuo et al., 2009; Guiver & Snelson, 2009) define ordering probabilities through Plackett–Luce-style sequential factorizations (Luce et al., 1959; Plackett, 1975), selecting items one at a time according to their target permutation. For example, ListMLE

defines the likelihood of a target permutation $y_1 \leq \cdots \leq y_n$ as

$$\mathbb{P}(y_1 \leq \cdots \leq y_n \mid \mathbf{s}) = \prod_{k=1}^{n} \frac{\exp(s_k)}{\sum_{j=1}^{k} \exp(s_j)},$$

where $\mathbf{s} = (s_1, \ldots, s_n)$ denotes the score vector. Such constructions retain an explicit full-order probability but introduce an implicit directionality tied to a particular sequential order. From an order-level perspective, ordering events are symmetric under order reversal: reversing the target ordering and predicted score ordering simultaneously should leave the learning problem unchanged. Sequential factorizations, however, generally break this order-reversal symmetry, as we discuss in Section 3.3.

## 3. Problem Formulation and Theoretical Motivation

In this section, we highlight two structural limitations in existing objectives: (i) surrogate losses via low-dimensional statistics do not uniquely determine the full-order event, and (ii) probabilistic models based on sequential factorizations introduce order-reversal asymmetries.

### 3.1. Standard Learning-to-Rank Setup

**Data and scoring model.** We consider a supervised learning-to-rank setting in which ranking instances are drawn from an underlying distribution $\mathcal{P}$ on $\mathcal{X} \times \mathcal{Y}$. A training dataset of size $N$ is

$$\mathcal{D} = \big\{ (\mathbf{x}^{(j)}, \mathbf{y}^{(j)}) \big\}_{j=1}^{N}, \qquad (\mathbf{x}^{(j)}, \mathbf{y}^{(j)}) \overset{\text{i.i.d.}}{\sim} \mathcal{P}.$$

For a single ranking instance, we write

$$\mathbf{x} = (x_1, \ldots, x_n), \qquad \mathbf{y} = (y_1, \ldots, y_n),$$

where $n$ is the list length, $x_i \in \mathbb{R}^d$ is the feature vector of item $i$, and $\mathbf{y}$ specifies the relative relevance of the $n$ items.

A ranking model $f_\theta$ maps each item to a real-valued score:

$$f_\theta : \mathbb{R}^d \to \mathbb{R}, \quad s_i = f_\theta(x_i), \quad \mathbf{s} = (s_1, \ldots, s_n) \in \mathbb{R}^n.$$

The goal is to learn $\theta$ so that the ordering induced by $\mathbf{s}$ is as consistent as possible with the ordering implied by $\mathbf{y}$.

Since the numerical values of $\mathbf{y}$ are only used to specify an ordering relation, we denote by

$$\pi_{\mathbf{y}} = \pi(\mathbf{y})$$

the permutation of $\{1, \ldots, n\}$ such that

$$y_{\pi(\mathbf{y})_1} \leq y_{\pi(\mathbf{y})_2} \leq \cdots \leq y_{\pi(\mathbf{y})_n},$$

where ties, if present, are handled separately in Appendix F.

The training loss for one instance is written as

$$\mathcal{L}(\theta; \mathbf{x}, \mathbf{y}) = F\big(\mathbf{s}, \pi(\mathbf{y})\big), \qquad \mathbf{s} = \big( f_\theta(x_1), \ldots, f_\theta(x_n) \big).$$

Here, $\mathcal{L}$ denotes the full loss as a function of the model parameters and the data, while $F$ denotes the same objective viewed in score space. For the order-level objectives considered in this paper, $F$ depends on the ordering structure $\pi(\mathbf{y})$, rather than on the raw numerical magnitudes of the labels.

**Ranking metrics.** Ranking quality is measured by comparing the predicted order $\pi(\mathbf{s})$ with the label order $\pi(\mathbf{y})$: exact permutation accuracy

$$\text{ACC} = \mathbb{E}\big[ \mathbf{1}\{\pi(\mathbf{s}) = \pi(\mathbf{y})\} \big],$$

Kendall's $\tau$ (the fraction of concordant pairs), and NDCG (emphasizing top-ranked items). These metrics are used for evaluation but are hard to optimize directly, being discontinuous in the scores.

**Surrogate objectives.** A common workaround optimizes differentiable surrogates instead: pairwise losses such as RankNet (Burges et al., 2005), listwise reweightings such as LambdaRank (Burges et al., 2006), and differentiable sorting methods (Grover et al., 2019; Petersen et al., 2022). These are efficient but abandon full-order probabilities, a limitation we revisit in Section 3.2.

**Order-level objective.** A more principled, maximum-likelihood alternative regards the score vector as random and directly models the probability of the observed ordering event. Treating $\mathbf{s} = (s_1, \ldots, s_n)$ as a random vector, the ideal order-level objective for an instance with label order $\pi(\mathbf{y})$ is

$$\mathcal{L}(\theta; \mathbf{x}, \mathbf{y}) = \log \mathbb{P}_\theta\big( s_{\pi(\mathbf{y})_1} \leq s_{\pi(\mathbf{y})_2} \leq \cdots \leq s_{\pi(\mathbf{y})_n} \big),$$

i.e., one maximizes the likelihood of the full-order event implied by the labels rather than a pairwise or position-wise surrogate. Representative instantiations include Plackett–Luce-style factorizations (ListNet (Cao et al., 2007), ListMLE (Xia et al., 2008)) that select items one at a time; Section 3.3 analyzes their structural directionality.

### 3.2. Limitation of Surrogate Objectives

A common strategy for designing the objective is through surrogates defined on lower-order statistics. Despite their algorithmic diversity, many such methods share a common structural form in which labels and scores enter through separate finite-dimensional representations:

$$\mathcal{L}(\theta; \mathbf{x}, \mathbf{y}) = F(\mathbf{s}, \pi(\mathbf{y})) = \langle \mathbf{G}(\pi_{\mathbf{y}}), \mathbf{H}(\mathbf{s}) \rangle, \qquad (1)$$

where $\mathbf{G}(\pi_{\mathbf{y}}) \in \mathbb{R}^d$ is a representation induced by the target permutation $\pi_{\mathbf{y}}$. $\mathbf{H}(\mathbf{s}) \in \mathbb{R}^d$ is a score-dependent representation produced by the model. $\langle \cdot, \cdot \rangle$ denotes the standard inner product (matrix-valued $\mathbf{G}$ and $\mathbf{H}$, as in Table 1, are read in vectorized form). The feature dimension $d$ is typically $O(n^2)$ or smaller for standard pairwise and listwise surrogates, and in particular $d \ll n!$.

Table 1 instantiates this abstraction for several representative ranking surrogates, showing how they fit into Equation (1).

| Method | $\mathbf{G}(\pi_{\mathbf{y}})$ | $\mathbf{H}(\mathbf{s})$ |
|---|---|---|
| RankNet | $\left[\mathbb{I}(y_i \geq y_j)\right]_{i,j}$ | $\left[-\log \sigma(s_i - s_j)\right]_{i,j}$ |
| LambdaRank | $\left[\Delta \mathrm{NDCG}_{ij}\right]_{i,j}$ | $\left[-\log \sigma(s_i - s_j)\right]_{i,j}$ |
| NeuralSort | $\mathbf{\Pi}(\mathbf{y})$ | $-\log \hat{\mathbf{\Pi}}(\mathbf{s})$ |

*Table 1.* Representative ranking surrogates under the separable formulation in Equation (1).

Equation (1) represents a wide range of existing approaches. Pairwise methods encode $\mathbf{G}(\pi_{\mathbf{y}})$ using pairwise comparison indicators; LambdaRank and related objectives reweight such comparisons by position-dependent gains; differentiable sorting methods compare soft permutation matrices $\hat{\mathbf{\Pi}}(\mathbf{s})$ against label-induced permutation matrices $\mathbf{\Pi}(\mathbf{y})$. In all cases, the supervision signal is compressed into a low-dimensional surrogate representation $\mathbf{G}(\pi_{\mathbf{y}})$ with $d \ll n!$.

To formalize the resulting limitation, consider a random permutation $\pi$ drawn from a distribution $p$ over full-order outcomes, and define the expected gradient field

$$\mathbf{g}_p(\mathbf{s}) = \mathbb{E}_{\pi \sim p}\left[\nabla_{\mathbf{s}} \mathcal{L}(\theta; \mathbf{x}, \mathbf{y})\right] = \mathbb{E}_{\pi \sim p}\left[\nabla_{\mathbf{s}} F(\mathbf{s}, \pi(\mathbf{y}))\right]. \tag{2}$$

For objectives of the form Equation (1), the expected gradient $\mathbf{g}_p(\mathbf{s})$ can only depend on the distribution $p$ through the finite-dimensional surrogate signal encoded by $\mathbf{G}(\pi)$. Different full-order label distributions that induce the same surrogate statistics therefore produce identical gradient fields.

**Definition 3.1** (Gradient separability). We say that a learning-to-rank objective is *gradient-separable* with respect to full-order distributions if for any two distributions $p_1 \neq p_2$ over permutations, there exists a score vector $\mathbf{s}$ such that $\mathbf{g}_{p_1}(\mathbf{s}) \neq \mathbf{g}_{p_2}(\mathbf{s})$.

Intuitively, gradient separability requires that different full-order label distributions induce distinguishable expected gradient fields somewhere in the score space, so that training can, in principle, respond differently to them.

**Proposition 3.2.** *Let $\mathbf{G}(\cdot) \in \mathbb{R}^d$ with $d < n! - 1$. Any objective of the form Equation (1) is not gradient-separable with respect to full-order distributions.*

The proof exploits the fact that the simplex of distributions over $n!$ permutations has dimension $n! - 1$, while any learn-

ing signal extracted through a lower dimensional representation $\mathbf{G}(\pi) \in \mathbb{R}^d$ necessarily collapses distinct full-order distributions. A constructive example based on pairwise statistics is given in Appendix C.

### 3.3. Limitation of Sequential Probabilistic Models

The second structural limitation concerns the directionality of sequential probabilistic models. Ordering events describe relative arrangements of items and should not depend on a particular choice of "forward" versus "backward" generation order. This suggests a natural symmetry property that full-order learning-to-rank objectives ought to satisfy.

**Definition 3.3** (Order-Reversal Invariance). A learning-to-rank objective $\mathcal{L}(\theta; \mathbf{x}, \mathbf{y}) = F(\mathbf{s}, \pi(\mathbf{y}))$ is said to satisfy *order-reversal invariance* if it is invariant under simultaneously reversing the orders implied by the labels and the scores via negation, i.e.,

$$F(\mathbf{s}, \pi(\mathbf{y})) = F(-\mathbf{s}, \pi(-\mathbf{y})) \qquad \text{for all } (\mathbf{s}, \mathbf{y}). \tag{3}$$

Note that negation reverses the induced permutation: if $\pi(\mathbf{y})$ is the permutation that sorts labels in ascending order, then $\pi(-\mathbf{y})$ is the reversed permutation of $\pi(\mathbf{y})$ (when there are no ties). An objective that depends only on the underlying ordering event, rather than on an arbitrary direction of generation, should therefore satisfy order-reversal invariance.

**Proposition 3.4** (Asymmetry of Plackett–Luce Models). *For $n \geq 3$, Plackett–Luce and related sequential factorization objectives do not satisfy order-reversal invariance.*

PL-style sequential factorization models define ordering probabilities through ordered selection procedures, introducing an intrinsic directionality into the learning objective, and bringing the asymmetry under order reversal operation. Formal proofs are presented in Appendix D, and experimental validations are presented in Appendix L.

## 4. Full-Order Probability Modeling

In this section we return to an order-level probabilistic perspective and let the deterministic outputs of $f_\theta$ parameterize a latent continuous model over relevance scores. This allows us to associate each ranking instance with a full-order probability for its observed ordering event. We first specify the latent score model and the corresponding full-order probability, and then refine the ordering event via auxiliary cut points to obtain the Full-Order Bound (FOB), a tractable objective that preserves the symmetry of ordering events and admits efficient optimization.

### 4.1. Latent Score Model

We begin with the case in which all labels within a ranking instance are strictly ordered (the extension to ties is

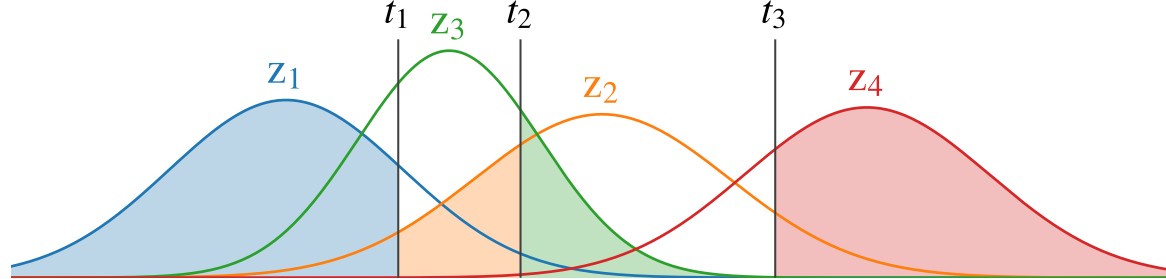

*Figure 1.* Illustration of the Full-Order Bound (FOB) for four latent scores. The cut points $t_1 \leq t_2 \leq t_3$ partition the real line into four ordered intervals. For the ordering event $z_1 \leq z_2 \leq z_3 \leq z_4$, FOB refines the event by assigning each latent score to its corresponding interval. This refined event is contained in the full-order event, and its probability factorizes across items, yielding a tractable lower bound.

discussed later in Appendix F). For a single instance with feature vectors $\{x_1, \ldots, x_n\}$ and labels $\{y_1, \ldots, y_n\}$, let $\pi(\mathbf{y})$ denote the permutation induced by the labels and, without loss of generality, reindex items so that $\pi_{\mathbf{y}}$ is the identity permutation and $y_1 < \cdots < y_n$. The observed ordering then corresponds to the index order $1, \ldots, n$.

To connect this setup to full-order probabilities, we associate each item $x_i$ with a latent random variable $z_i$ representing its (unobserved) relevance, and write $\mathbf{z} = (z_1, \ldots, z_n)$ for the latent score vector. Here $z_i$ instantiates the random-score viewpoint of Section 3.1, replacing the abstract random score by an explicit latent variable parameterized by $f_\theta$. In our implementation we take the latent scores to follow a heteroscedastic (or homoscedastic) Gaussian model

$$z_i = \mu_i + \sigma_i \varepsilon_i, \quad (\mu_i, \sigma_i) = f_\theta(x_i), \quad \varepsilon_i \overset{\text{i.i.d.}}{\sim} \mathcal{N}(0, 1). \tag{4}$$

so that $f_\theta$ now outputs a location and a scale parameter for each item, extending the scalar score $s_i$ of Section 3.1 to the parameters of a latent distribution; the location $\mu_i$ recovers a deterministic ranking score and is used as such at inference time. The construction of FOB only requires a continuous symmetric location–scale family, so the Gaussian choice in Equation (4) can be replaced by alternatives such as logistic or Laplace distributions. In particular, when the latent base density is log-concave (e.g., Gaussian, Laplace, logistic), the inner tightening problem over cut points is concave; we formalize this in Theorem 5.1.

As shown in Equation (4), all cross-item dependence is mediated through the shared conditioning on the feature vectors $x_i$ and the model parameters $\theta$, whereas the residual noise variables $\varepsilon_i$ are mutually independent.

Under this latent score model, the observed ordering corresponds to the event

$$\mathcal{E} = \{z_1 \leq \cdots \leq z_n\}, \tag{5}$$

a single ordering cone in the continuous latent space. Under the continuous density, $\leq$ and $<$ give the same event up to a null set. The ideal order-level objective would maximize the

log-likelihood of the full-order event $\log \mathbb{P}_\theta(\mathcal{E})$ as induced by the joint distribution of $\mathbf{z}$ predicted by $f_\theta$.

For $n = 2$, $\mathbb{P}_\theta(\mathcal{E})$ admits simple closed-form expressions under Equation (4), but for $n \geq 3$ it becomes a high-dimensional orthant (or cone) probability with no simple closed form, and direct numerical integration is impractical at learning-to-rank scales. Rather than reverting to low-dimensional surrogates or directional factorizations, we approximate $\log \mathbb{P}_\theta(\mathcal{E})$ by a tractable lower bound constructed from a refined subset of the ordering event.

### 4.2. Interval-Based Construction of the Full-Order Bound

To obtain a tractable lower bound on $\log \mathbb{P}_\theta(\mathcal{E})$, we refine the ordering event $\mathcal{E}$ in Equation (5) using shared cut points. Introduce ordered cut points $\mathbf{t} = (t_1, \ldots, t_{n-1})$ satisfying $t_1 \leq \cdots \leq t_{n-1}$, and set $t_0 = -\infty$, $t_n = +\infty$. These cut points partition the real line into $n$ intervals $(t_{i-1}, t_i)$. The refined event

$$\mathcal{E}(\mathbf{t}) = \{ t_{i-1} \leq z_i \leq t_i, \ i = 1, \ldots, n \}$$

requires each latent score $z_i$ to fall into its designated interval (Figure 1 illustrates this construction for $n = 4$). Since $\mathcal{E}(\mathbf{t}) \subseteq \mathcal{E}$ by construction, we obtain

$$\mathbb{P}_\theta(\mathcal{E}) \geq \mathbb{P}_\theta(\mathcal{E}(\mathbf{t})), \quad \forall \mathbf{t} \in \mathcal{T},$$

where $\mathcal{T} = \{ \mathbf{t} : t_1 \leq \cdots \leq t_{n-1} \}$ is the feasible set.

Under the independence assumption in Section 4.1, we define the Full-Order Bound

$$\text{FOB}(\mathbf{t}) \triangleq \log \mathbb{P}_\theta(\mathcal{E}(\mathbf{t})) = \sum_{i=1}^n \log \mathbb{P}_\theta(t_{i-1} \leq z_i \leq t_i), \tag{6}$$

which is a lower bound on $\log \mathbb{P}_\theta(\mathcal{E})$ for $\mathbf{t} \in \mathcal{T}$.

For the Gaussian latent score model in Equation (4), each interval probability $\mathbb{P}_\theta(t_{i-1} \leq z_i \leq t_i)$ admits a closed-form expression in terms of the standard normal cumulative

---

**Algorithm 1** Optimization of the Full-Order Bound (FOB) with Safe-Region Gradient Ascent (SRGA).

| **Main training loop** | **Safe-Region Gradient Ascent (SRGA)** |
|---|---|
| **Require:** Items $(x_1, \ldots, x_n)$, labels $(y_1 \leq \cdots \leq y_n)$ | **Require:** Current cut points $\mathbf{t} = (t_1 \leq \cdots \leq t_{n-1})$, |
| 1: Compute $(\mu_i, \sigma_i) = f_\theta(x_i)$ for $i = 1, \ldots, n$ |      parameters $(\mu_i, \sigma_i)$ |
| 2: Normalization via Equation (8) and Equation (9) | 1: Set $t_0 = -\infty$, $t_n = +\infty$ |
| 3: Initialize $t_i$ (e.g., as midpoints of adjacent $\mu_i$) | 2: Compute probabilities $p_i = \mathbb{P}(t_{i-1} \leq z_i \leq t_i)$ |
| 4: **for** $k = 1$ to $K$ **do** | 3: Clip $p_i$ to $[\varepsilon, 1 - \varepsilon]$ |
| 5:     $\mathbf{t} \leftarrow \text{SRGA}(\mathbf{t}, \mu, \sigma)$ | 4: Compute analytic gradients $\nabla_{\mathbf{t}}\text{FOB}$ |
| 6: **end for** | 5: Apply a gradient step on $\mathbf{t}$ and clamp each $t_i$ to the |
| 7: Compute $\nabla_\theta \text{FOB}(\mathbf{t})$ and update $\theta$ |      interval in Equation (11) |

---

distribution function. Consequently, Equation (6) yields a tractable lower bound on the full-order probability.

The tightest instance of the Full-Order Bound for fixed model parameters is obtained by maximizing $\text{FOB}(\mathbf{t})$ over the feasible set $\mathcal{T} = \{\mathbf{t} : t_1 \leq \cdots \leq t_{n-1}\}$. The resulting learning objective is

$$\max_\theta \max_{\mathbf{t} \in \mathcal{T}} \text{FOB}(\mathbf{t}; \theta),$$

where $\theta$ parameterizes the latent score model $f_\theta$. In practice, the inner maximization is performed approximately using a small number of constrained updates, providing a tight and stable training signal. The optimization procedure is detailed in Section 5.

**Order-level objective and its symmetry.** Recall from Section 3.1 that the ideal order-level objective treats the score as random; the latent model of Section 4.1 realizes this through the predicted parameters $(\boldsymbol{\mu}, \boldsymbol{\sigma}) = f_\theta(\mathbf{x})$, which determine the law of $\mathbf{z}$. Writing the tightened bound as an order-level objective in the abstract $F(\cdot, \pi(\mathbf{y}))$ form of Section 3.1,

$$F_{\text{FOB}}(\boldsymbol{\mu}, \boldsymbol{\sigma}; \pi(\mathbf{y})) \triangleq -\max_{\mathbf{t} \in \mathcal{T}} \text{FOB}(\mathbf{t}; \theta), \qquad (7)$$

a deterministic functional of the predicted latent-score parameters $(\boldsymbol{\mu}, \boldsymbol{\sigma})$ and the label-induced ordering $\pi(\mathbf{y})$ ($\mathbf{z}$ is integrated out inside FOB, and $\pi(\mathbf{y})$ enters via the reindexing of Section 4.1 that matches the $i$-th interval to the $i$-th label-ranked item). This is exactly the per-instance training loss used in Section 5.4.

**Conjecture 4.1** (Full linear rank of the Gaussian FOB loss). *For independent heteroscedastic Gaussian latents $z_i \sim \mathcal{N}(\mu_i, \sigma_i^2)$ and $n \geq 2$, the loss family $\{F_{\text{FOB}}(\boldsymbol{\mu}, \boldsymbol{\sigma}; \pi) : \pi \in S_n\}$ is linearly independent on $\mathbb{R}^n \times (0, \infty)^n$. Consequently, the tightened FOB loss has full linear rank $n!$ over permutations.*

Conjecture 4.1 would rule out any exact lower-dimensional linear permutation representation of the tightened Gaussian FOB objective. We do not rely on this statement as

a theorem; Appendix B provides numerical evidence for the conjecture on small list sizes, with the orbit evaluation matrices attaining full numerical rank for all tested cases up to $n = 6$.

When the score is modeled as a latent distribution, the deterministic score of Definition 3.3 is replaced by its location $\boldsymbol{\mu}$, while the scale $\boldsymbol{\sigma}$ is a sign-invariant dispersion. Order reversal therefore acts on the parameters through $\mathcal{R} : (\mu_i, \sigma_i) \mapsto (-\mu_{n+1-i}, \sigma_{n+1-i})$, together with the label reversal $\pi(\mathbf{y}) \mapsto \pi(-\mathbf{y})$.

**Proposition 4.2** (Order-Reversal Invariance of FOB). *For any symmetric base density $p_0$ (e.g., Gaussian, logistic, Laplace) and all $(\boldsymbol{\mu}, \boldsymbol{\sigma}, \mathbf{y})$, $F_{\text{FOB}}$ satisfies Definition 3.3:*

$$F_{\text{FOB}}(\boldsymbol{\mu}, \boldsymbol{\sigma}; \pi(\mathbf{y})) = F_{\text{FOB}}(\mathcal{R}(\boldsymbol{\mu}, \boldsymbol{\sigma}); \pi(-\mathbf{y})).$$

Unlike the directional Plackett–Luce factorization of Proposition 3.4, FOB is built directly from the ordering event through a cut-point construction that is symmetric under negation; the proof is given in Appendix D.2.

## 5. Optimization of the FOB

This section describes how we optimize the Full-Order Bound (FOB) introduced in Section 4. The objective depends on both the model parameters $\theta$, through the latent score statistics $(\mu_i, \sigma_i)$, and the auxiliary cut points $\mathbf{t} = (t_1, \ldots, t_{n-1})$. We adopt a bilevel perspective: for each training instance, we first tighten the bound by approximately maximizing it with respect to $\mathbf{t}$, and then update $\theta$ using the resulting tightened objective.

### 5.1. Normalization and Invariances

The Full-Order Bound is invariant under positive affine transformations of the latent score space. For any $a > 0$ and $b \in \mathbb{R}$, applying $z_i \mapsto az_i + b$ and $t_i \mapsto at_i + b$ leaves both the ordering event and the value of $\text{FOB}(\mathbf{t})$ unchanged. Thus the objective is insensitive to the global location and scale of the latent scores. If not properly constrained, these redundant degrees of freedom can lead to ill-conditioned optimization.

To remove this redundancy, we normalize the predicted latent score parameters within each list. Let $\mu_i$ and $\sigma_i$ denote the mean and scale of the latent score $z_i$. We compute the empirical mean and standard deviation of the list-level (or batch-level) latent score mixture:

$$\bar{\mu} = \frac{1}{n} \sum_{i=1}^{n} \mu_i, \qquad \bar{\sigma} = \sqrt{\frac{1}{n} \sum_{i=1}^{n} \left( \mu_i^2 + \sigma_i^2 \right) - \bar{\mu}^2}. \quad (8)$$

We then apply the normalization

$$\mu_i \leftarrow \frac{\mu_i - \bar{\mu}}{\bar{\sigma}}, \qquad \sigma_i \leftarrow \frac{\sigma_i}{\bar{\sigma}}. \quad (9)$$

This transformation preserves all ordering probabilities while fixing the global location and scale, and in practice leads to more stable optimization of FOB with minimal additional computation.

### 5.2. Concave Inner Problem

For fixed latent score parameters $(\mu_i, \sigma_i)_{i=1}^{n}$ with $\sigma_i > 0$, the inner problem is to maximize the Full-Order Bound defined in Equation (6) over the feasible set $\mathcal{T}$.

**Theorem 5.1** (Concavity of the FOB inner problem). *For independent $z_i$ with log-concave density functions, $\mathrm{FOB}(\mathbf{t})$ is concave over $\mathcal{T}$. Consequently, the inner problem*

$$\max_{\mathbf{t} \in \mathcal{T}} \mathrm{FOB}(\mathbf{t}) \quad (10)$$

*is a convex optimization problem.*

The proof is provided in Appendix E. This result guarantees that tightening the bound with respect to $\mathbf{t}$ does not introduce spurious local optima and can, in principle, be solved reliably with first-order methods. In practice, a small number of inner updates is sufficient to obtain a tight bound for updating the model parameters.

### 5.3. Safe-Region Gradient Ascent

The main practical challenge in optimizing Equation (10) is the constraint $\mathbf{t} \in \mathcal{T}$. Barrier or projection methods can require careful tuning or produce unstable zigzag updates near narrow feasible regions. We instead adopt a *safe-region* gradient ascent (SRGA) scheme that enforces the ordering constraint at every update. The key idea is to restrict each cut point $t_i$ to a local interval determined by its neighbors, so that an unconstrained gradient step followed by clamping automatically preserves $\mathbf{t} \in \mathcal{T}$.

Concretely, for each $t_i$ we define the safe region

$$\mathcal{I}_i = \left\{ (1 - \lambda)u + \lambda t_i \mid u \in [t_{i-1}, t_{i+1}] \right\}, \quad (11)$$

where $\lambda \in (0.5, 1)$. Since the objective contains terms of the form $\log \mathbb{P}(t_i \leq z \leq t_{i+1})$, any feasible solution satisfies

$t_i \leq t_{i+1}$, under which adjacent safe regions $\mathcal{I}_i$ and $\mathcal{I}_{i+1}$ are non-overlapping.

In implementation, we first apply a gradient step on $t_i$ as if the problem were unconstrained, and then clamp the updated value back to $\mathcal{I}_i$ in Equation (11). By construction, the resulting $\mathbf{t}$ remains in $\mathcal{T} = \{\mathbf{t} : t_1 \leq \cdots \leq t_{n-1}\}$ at every SRGA step, without requiring explicit projection onto the global feasible set.

This lets us tighten with analytic derivatives of $\mathrm{FOB}(\mathbf{t})$ and a simple clamp—no barriers or penalties—while confining each $t_i$ to a small neighborhood between its neighbors, giving a smoother trajectory than aggressive projected methods and stable tightening in practice. The overall procedure is summarized in Algorithm 1. Since there are $n - 1$ cut points, each inner step costs $O(n)$ per instance, and the overall overhead remains modest for small numbers of inner iterations. An extension of the inner tightening to tied labels is described in Appendix F. We also study sensitivity of SRGA hyperparameters in Appendix O.

### 5.4. Final Training Objective and Loss Properties

We cast training as a bilevel optimization problem that maximizes a tightened lower bound on the full-order probability. For each instance, we first compute latent parameters $(\mu_i, \sigma_i) = f_\theta(x_i)$, normalize them according to Equations (8) and (9), and then tighten the FOB by solving

$$\mathbf{t}^* = \arg \max_{\mathbf{t} \in \mathcal{T}} \mathrm{FOB}(\mathbf{t}; \theta).$$

In practice, $\mathbf{t}^*$ is approximated by a fixed number of SRGA steps, and we do not differentiate through the inner iterations, avoiding costly second-order terms.

The final training loss for a single instance is defined as the negative tightened bound:

$$\mathcal{L}(\theta; \mathbf{x}, \mathbf{y}) \triangleq - \mathrm{FOB}(\mathbf{t}^*; \theta)$$

$$= - \sum_{i=1}^{n} \log \mathbb{P}_\theta \left( t_{i-1}^* \leq z_i \leq t_i^* \right).$$

Minimizing $\mathcal{L}$ with respect to $\theta$ is therefore equivalent to maximizing a valid lower bound on the full-order probability of the observed ordering event for each instance.

Directly optimizing the full-order event $\{z_1 \leq \cdots \leq z_n\}$ from the start can, however, be numerically unstable: early in training, the model may place most mass on very sharp regions of the latent space, leading to extremely small interval probabilities and exploding gradients. To mitigate this, we employ a simple list-length warmup scheme. During an initial warmup phase, each training step operates on a random sublist of length $m$, where $m$ increases linearly from 4 to the full list length $n$. Sublists are formed by sampling $m$ indices uniformly without replacement from $\{1, \ldots, n\}$.

*Table 2.* Permutation accuracy (ACC, %) on the MNIST permutation task with different list lengths. Results are reported as mean $\pm$ 95% confidence intervals over five runs.

|  | $n=4$ | $n=6$ | $n=8$ | $n=16$ | $n=32$ |
|---|---|---|---|---|---|
| RankNet | $85.52 \pm 0.94$ | $68.54 \pm 6.07$ | $47.88 \pm 6.84$ | $3.34 \pm 1.96$ | $0.0 \pm 0.0$ |
| LambdaRank | $85.34 \pm 1.79$ | $67.16 \pm 4.96$ | $44.13 \pm 9.83$ | $3.24 \pm 1.36$ | $0.0 \pm 0.0$ |
| ListMLE | $82.59 \pm 0.62$ | $58.57 \pm 2.65$ | $40.05 \pm 3.70$ | $2.92 \pm 0.88$ | $0.0 \pm 0.0$ |
| Gumbel Sinkhorn | $67.88 \pm 0.86$ | $59.18 \pm 1.04$ | $33.46 \pm 2.79$ | $2.37 \pm 0.61$ | $0.0 \pm 0.0$ |
| NeuralSort | $75.66 \pm 1.54$ | $47.66 \pm 1.21$ | $24.18 \pm 1.28$ | $0.16 \pm 0.06$ | $0.0 \pm 0.0$ |
| DiffSort | $89.41 \pm 0.40$ | $78.51 \pm 1.91$ | $66.54 \pm 1.02$ | $25.28 \pm 3.44$ | $0.23 \pm 0.37$ |
| EF-DSF | $89.66 \pm 0.19$ | $79.16 \pm 0.86$ | $68.13 \pm 0.88$ | $26.51 \pm 2.57$ | $0.45 \pm 0.49$ |
| FOB | $\mathbf{89.88} \pm 0.78$ | $\mathbf{79.50} \pm 0.44$ | $\mathbf{68.20} \pm 1.61$ | $\mathbf{27.55} \pm 2.38$ | $\mathbf{1.25} \pm 0.81$ |

This keeps the ordering constraints less extreme in the early stages, preventing gradient explosion while allowing the model to gradually adapt to longer lists. An ablation study of this warmup strategy is reported in Appendix I.

The training loss is the order-level objective $F_{\text{FOB}}(\boldsymbol{\mu}, \boldsymbol{\sigma}; \pi(\mathbf{y}))$ of Equation (7), with the inner maximization approximated by SRGA. It therefore inherits the two structural properties targeted in this work: it is derived from the full-order event rather than a low-dimensional surrogate; Conjecture 4.1 and the numerical evidence in Appendix B further support the view that FOB retains high-dimensional permutation structure. It is also order-reversal invariant (Proposition 4.2), in contrast to both surrogate and sequential probabilistic losses. These properties contribute to its empirical robustness in the experiments reported in Section 6.

## 6. Experiments

We evaluate the proposed method on a synthetic permutation task built from MNIST (LeCun et al., 1998) dataset and on two standard large-scale learning-to-rank benchmarks, Web30K and Istella. Our experiments aim to answer: (i) whether directly optimizing a full-order probability via FOB improves ranking performance compared with pairwise, listwise, differentiable-sorting, and sequential factorization objectives, and (ii) whether the proposed optimization procedure remains stable and efficient in practice. Unless stated otherwise, we present results as the mean over five random seeds for the synthetic MNIST permutation task and ten random seeds for the Web30K and Istella datasets. In all cases, we report 95% confidence intervals based on the t-distribution. Additional training details and hyperparameters are provided in Appendix H.

### 6.1. MNIST Permutation Task

We first consider a synthetic ranking task constructed from MNIST dataset, following prior work on differentiable sort-

ing (Petersen et al., 2022). Each training instance consists of a list of $n$ composite images, where each image contains four horizontally concatenated MNIST digits representing a 4-digit number. The ground-truth label for each image is the numeric value induced by the digits, so all items in a list are fully comparable. We evaluate settings with $n \in \{4, 6, 8, 16, 32\}$ to study the effect of list length on different ranking objectives. Performance is measured using exact permutation accuracy (ACC).

All methods share the same convolutional backbone followed by a small MLP head for rank score prediction. Baseline methods cover representative paradigms: RankNet (Burges et al., 2005) and LambdaRank (Burges et al., 2006) as pairwise and listwise surrogates, ListMLE (Xia et al., 2008) as a PL-style sequential factorization method, and Gumbel-Sinkhorn (Mena et al., 2018), NeuralSort (Grover et al., 2019), DiffSort (Petersen et al., 2022), and EF-DSF (Kim et al., 2024) as differentiable sorting methods.

For the proposed FOB objective, we reuse the same backbone and add a lightweight auxiliary head to predict the latent scale $\sigma_i$, while the mean $\mu_i$ serves as the deterministic ranking score at inference time. Full architectures and hyperparameters are summarized in Appendix H.

As shown in Table 2, directly optimizing a lower bound on the full-order probability yields consistently higher ACC across all list lengths. The performance gap between FOB and baseline methods widens as $n$ increases, suggesting that these surrogate objectives may become increasingly misaligned with the exact ordering metric in this controlled setting, whereas FOB continues to capture full-order structure. Note that exact permutation accuracy is an extremely stringent metric, especially for large $n$: for $n=32$, a single error in the induced ordering renders the entire list incorrect.

### 6.2. Web30K and Istella Benchmarks

We next conduct experiments on two standard large-scale learning-to-rank benchmarks, Web30K (Qin & Liu, 2013)

*Table 3.* NDCG and Kendall's $\tau$ results (%) on the Web30K and Istella benchmarks. Mean $\pm$ 95% CI are reported over ten runs.

| Method | Web30K | | | Istella | | |
|---|---|---|---|---|---|---|
| | N@10 | N@20 | Kendall's $\tau$ | N@10 | N@20 | Kendall's $\tau$ |
| RankNet | $50.28 \pm 0.23$ | $54.46 \pm 0.21$ | $34.53 \pm 0.18$ | $73.34 \pm 0.21$ | $79.95 \pm 0.17$ | $85.96 \pm 0.09$ |
| PairwiseHinge | $50.20 \pm 0.19$ | $54.41 \pm 0.16$ | $\mathbf{34.59} \pm 0.14$ | $73.22 \pm 0.25$ | $79.81 \pm 0.21$ | $85.91 \pm 0.10$ |
| LambdaRank | $51.22 \pm 0.16$ | $55.30 \pm 0.13$ | $33.94 \pm 0.17$ | $73.88 \pm 0.13$ | $80.34 \pm 0.11$ | $85.69 \pm 0.13$ |
| ApproxNDCG | $49.24 \pm 0.36$ | $53.28 \pm 0.27$ | $31.38 \pm 0.24$ | $71.31 \pm 0.17$ | $78.16 \pm 0.15$ | $82.90 \pm 0.22$ |
| ListMLE | $49.66 \pm 0.13$ | $53.81 \pm 0.13$ | $33.66 \pm 0.21$ | $71.12 \pm 0.42$ | $77.83 \pm 0.36$ | $83.90 \pm 0.35$ |
| NeuralSort | $48.52 \pm 0.38$ | $52.90 \pm 0.34$ | $32.56 \pm 0.62$ | $71.60 \pm 0.29$ | $78.35 \pm 0.25$ | $84.64 \pm 0.21$ |
| DiffSort | $48.77 \pm 0.28$ | $53.03 \pm 0.23$ | $31.91 \pm 0.40$ | $72.22 \pm 0.27$ | $78.74 \pm 0.22$ | $84.13 \pm 0.30$ |
| EF-DSF | $48.94 \pm 0.24$ | $53.13 \pm 0.19$ | $32.08 \pm 0.24$ | $72.37 \pm 0.28$ | $78.91 \pm 0.28$ | $84.40 \pm 0.47$ |
| FOB | $51.11 \pm 0.16$ | $55.17 \pm 0.11$ | $34.05 \pm 0.22$ | $73.44 \pm 0.22$ | $79.99 \pm 0.18$ | $\mathbf{86.21} \pm 0.08$ |
| LAMBDAFOB | $\mathbf{51.28} \pm 0.15$ | $\mathbf{55.32} \pm 0.11$ | $33.98 \pm 0.19$ | $\mathbf{73.89} \pm 0.13$ | $\mathbf{80.35} \pm 0.12$ | $85.96 \pm 0.09$ |

and Istella (Dato et al., 2016). We use the conventional train/validation/test splits and set the list length to 64 by randomly sampling documents for each query. All results are averaged over 10 runs with different random seeds. We report NDCG@10 and NDCG@20, as well as Kendall's $\tau$ statistic, which measures the full list rank correlation based on concordant/discordant pairs.

Both datasets provide five-level graded labels, so many items within a query share ties. Accordingly, we use four cut points for FOB, independent of the list length, and optimize them using the aggregated tightening objective described in Appendix F. All methods share the same backbone network; full configurations (batch size, learning-rate schedule, and SRGA settings) are deferred to Appendix H.

On Web30K and Istella, we additionally compare against the pairwise hinge loss (Herbrich et al., 2000) and ApproxNDCG (Qin et al., 2010). Table 3 summarizes the results. Overall, FOB is strongest on full-list consistency: it matches the best baseline on Web30K within confidence intervals and attains the highest Kendall's $\tau$ on Istella, indicating improved pairwise ordering consistency.

For completeness, we include results for LAMBDAFOB, a metric-aligned variant of FOB used for comparisons with Lambda-style objectives, which emphasize high-ranked labels. As shown in Table 3, LAMBDAFOB recovers the expected NDCG gains while remaining comparable to FOB in Kendall's $\tau$. Details are provided in Appendix G.

### 6.3. Ablations and Additional Analyses

In our ablations, disabling any of the four components — latent score normalization, inner tightening over $\mathbf{t}$, safe-region updates, or list-length warmup — consistently degrades accuracy and stability, most severely for normalization or inner tightening (Appendix I). Performance improves rapidly with a few SRGA steps and then plateaus, so modest budgets

suffice (Appendix K).

FOB is effective across location–scale families, but log-concave choices (Gaussian, Laplace) optimize more stably than non-log-concave ones (Cauchy), with Gaussian best overall (Appendix J). Heteroscedastic scales outperform homoscedastic ones, so $\sigma_i$ captures useful uncertainty rather than redundant rescaling (Appendix P).

The optimized FOB lower bound tracks Monte Carlo estimates of the exact full-order probability throughout training for $n \in \{4, 8, 12\}$; it loosens as $n$ grows but follows the same trend, remaining a meaningful training signal (Appendix N). FOB is stable across SRGA hyperparameters (Appendix O) and, unlike PL-based ListMLE, insensitive to label reversal (Appendix L), empirically illustrating the practical implication of Proposition 4.2. SRGA tightening adds little overhead even with more inner steps (Appendix M), and FOB retains a large Kendall's $\tau$ margin over RankNet, ListMLE, and DiffSort at $n = 256, 512$ (Appendix Q).

## 7. Conclusion

We studied learning to rank from an order-level probabilistic viewpoint and identified two limitations of existing objectives: low-dimensional surrogates cannot distinguish full-order distributions at the gradient level (gradient-level non-separability), and Plackett–Luce-style sequential models violate order-reversal invariance. To address both, we introduced the Full-Order Bound (FOB), which targets full-order probabilities through shared cut points over latent scores and constructs a tractable lower bound on the ordering event. The resulting bilevel objective admits a convex inner tightening problem and can be optimized efficiently with stable training dynamics. Experiments on synthetic permutation tasks and large-scale benchmarks demonstrate competitive performance, with clear improvements on full-list ordering metrics in the settings studied here.

## Impact Statement

This paper presents work whose goal is to advance the field of machine learning. There are many potential societal consequences of our work, none of which we feel must be specifically highlighted here.

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

# A. Limitations of Surrogate Objectives

This appendix formalizes the information bottleneck induced by surrogate ranking objectives discussed in Section 3.2. Throughout, the loss is defined with respect to a label vector $\mathbf{y}$, and randomness is introduced only through distributions over the induced permutations $\pi_{\mathbf{y}}$.

## A.1. Unified Surrogate Formulation

We recall the unified surrogate formulation from Section 3.2. A broad class of ranking surrogates can be written in the abstract form

$$\mathcal{L}(\theta; \mathbf{x}, \mathbf{y}) = \langle \mathbf{G}(\pi_{\mathbf{y}}), \mathbf{H}(\mathbf{s}) \rangle, \tag{12}$$

where $\pi_{\mathbf{y}}$ denotes the permutation induced by the labels $\mathbf{y}$, $\mathbf{G}(\pi_{\mathbf{y}}) \in \mathbb{R}^d$ is a permutation-dependent supervision representation, and $\mathbf{H}(\mathbf{s}) \in \mathbb{R}^d$ is a score-dependent representation produced by the model. The inner product $\langle \cdot, \cdot \rangle$ is understood in the standard (vectorized) sense. For standard pairwise and listwise surrogates, the feature dimension $d$ is typically $O(n^2)$ or smaller, and in particular $d \ll n!$.

## A.2. Gradient-Level Non-Separability

Taking gradients of Equation (12) with respect to the score vector $\mathbf{s}$ yields

$$\nabla_{\mathbf{s}} \mathcal{L}(\theta; \mathbf{x}, \mathbf{y}) = \left( \nabla_{\mathbf{s}} \mathbf{H}(\mathbf{s}) \right)^{\top} \mathbf{G}(\pi_{\mathbf{y}}), \tag{13}$$

where $\nabla_{\mathbf{s}} \mathbf{H}(\mathbf{s})$ is the Jacobian of $\mathbf{H}$ with respect to $\mathbf{s}$.

Consider now a probability distribution $p$ over permutations $\pi \in S_n$. This induces a distribution over label vectors $\mathbf{y}$ through the mapping $\mathbf{y} \mapsto \pi_{\mathbf{y}}$. The expected gradient field associated with $p$ is defined as

$$\mathbf{g}_p(\mathbf{s}) = \mathbb{E}_{\pi \sim p} \left[ \nabla_{\mathbf{s}} \mathcal{L}(\theta; \mathbf{x}, \mathbf{y}) \right]. \tag{14}$$

Using Equation (13), we obtain

$$\mathbf{g}_p(\mathbf{s}) = \left( \nabla_{\mathbf{s}} \mathbf{H}(\mathbf{s}) \right)^{\top} \mathbb{E}_{\pi \sim p} \left[ \mathbf{G}(\pi) \right]. \tag{15}$$

Thus, for any fixed $\mathbf{s}$, the dependence of the expected gradient on $p$ is entirely through the surrogate moment $\mathbb{E}_{\pi \sim p}[\mathbf{G}(\pi)]$.

To formalize the resulting non-separability, consider the linear map

$$T : \Delta(S_n) \to \mathbb{R}^d, \qquad T(p) \triangleq \mathbb{E}_{\pi \sim p}[\mathbf{G}(\pi)],$$

where $\Delta(S_n)$ denotes the probability simplex over all $n!$ permutations. The domain $\Delta(S_n)$ has dimension $n! - 1$, while the range of $T$ lies in $\mathbb{R}^d$ with $d < n! - 1$. Since $T$ is linear in $p$ and maps a space of (affine) dimension $n! - 1$ into one of dimension at most $d$, the map $T$ cannot be injective whenever $d < n! - 1$. Hence, there exist distinct distributions $p_1 \neq p_2$ over permutations such that

$$\mathbb{E}_{\pi \sim p_1}[\mathbf{G}(\pi)] = \mathbb{E}_{\pi \sim p_2}[\mathbf{G}(\pi)]. \tag{16}$$

By Equation (15), these distributions induce identical expected gradient fields for all score vectors $\mathbf{s}$:

$$\mathbf{g}_{p_1}(\mathbf{s}) = \mathbf{g}_{p_2}(\mathbf{s}) \qquad \text{for all } \mathbf{s}. \tag{17}$$

Consequently, $p_1$ and $p_2$ are indistinguishable under gradient-based optimization, even though they may assign different probability mass to some full-order events. This gradient-level non-separability is intrinsic to the use of low-dimensional surrogate representations $\mathbf{G}(\pi)$ with $d \ll n!$ and cannot be removed by reweighting or modifying $\mathbf{H}(\mathbf{s})$ within the surrogate framework Equation (12). It directly implies the non-separability result stated in Proposition 3.2.

# B. Numerical Evidence for the Full-Rank Conjecture

This appendix records numerical evidence for Conjecture 4.1. The previous version of this appendix attempted to prove full rank via a multiscale determinant argument. We instead state the full-rank property as a conjecture and provide a

reproducible numerical verification protocol. The experiments below provide evidence for Conjecture 4.1, but do not constitute a mathematical proof for arbitrary $n$.

Let

$$\mathcal{T} = \{\mathbf{t} \in \mathbb{R}^{n-1} : t_1 \leq \cdots \leq t_{n-1}\}, \qquad t_0 = -\infty, \quad t_n = +\infty.$$

For a permutation $\pi = (\pi_1, \ldots, \pi_n)$, define the relabeled refined event

$$\mathcal{E}_\pi(\mathbf{t}) = \{t_{r-1} \leq z_{\pi_r} \leq t_r, \ r = 1, \ldots, n\}.$$

For independent Gaussian latents with parameters $(\boldsymbol{\mu}, \boldsymbol{\sigma})$, we use the shorthand

$$F_\pi(\boldsymbol{\mu}, \boldsymbol{\sigma}) = F_{\text{FOB}}(\boldsymbol{\mu}, \boldsymbol{\sigma}; \pi) = -\log \max_{\mathbf{t} \in \mathcal{T}} \mathbb{P}(\mathcal{E}_\pi(\mathbf{t})). \tag{18}$$

This is exactly the tightened order-level objective in Equation (7), written with the target permutation made explicit. Conjecture 4.1 asserts that the functions $\{F_\pi : \pi \in S_n\}$ are linearly independent on $\mathbb{R}^n \times (0, \infty)^n$.

**Orbit evaluation matrix.** To test this conjecture numerically, fix the identity-order base vector $\mathbf{a} = (1, \ldots, n)$, a scale $s > 0$, and a common standard deviation $\sigma > 0$. For each row permutation $\rho \in S_n$, define an orbit point by relabeling the scaled base means,

$$\mu_{\rho_k}^{\rho, s} = s\, a_k = s\, k, \qquad k = 1, \ldots, n,$$

and let $\boldsymbol{\sigma} = \sigma\mathbf{1}$. The item-relabeling equivariance of the interval construction gives the orbit evaluation matrix

$$M_s(\rho, \pi) = F_\pi(\boldsymbol{\mu}^{\rho, s}, \sigma\mathbf{1}) = f_s(\rho^{-1}\pi), \qquad f_s(\tau) \triangleq F_\tau(s\mathbf{a}, \sigma\mathbf{1}).$$

Here $\rho^{-1}\pi$ denotes the permutation obtained by applying the inverse item relabeling $\rho^{-1}$ to each item in the ordered list $\pi$. Hence $M_s$ is an $S_n$-group-circulant evaluation matrix. The orbit construction reduces the number of inner optimizations from $(n!)^2$ to $n!$: it suffices to compute $f_s(\tau)$ for every $\tau \in S_n$, and then fill $M_s(\rho, \pi) = f_s(\rho^{-1}\pi)$.

**Numerical protocol.** For each $\tau \in S_n$, we numerically maximize

$$\sum_{r=1}^n \log \mathbb{P}(t_{r-1} \leq z_{\tau_r} \leq t_r) \quad \text{over} \quad \mathbf{t} \in \mathcal{T},$$

with $z_i \sim \mathcal{N}(si, \sigma^2)$. Gaussian interval probabilities are evaluated using stable normal CDF routines. The constrained inner problem is a concave maximization over the convex cone $\mathcal{T}$ for Gaussian latents, and the implementation still uses multiple initializations to reduce numerical sensitivity. After constructing $M_s$, we compute its singular values, numerical rank, smallest singular value, largest singular value, and condition number.

The table below reports runs with $\sigma = 1$, $s \in \{0.5, 1.0, 2.0\}$, and $n \in \{3, 4, 5, 6\}$. The numerical rank is computed from the singular values using the standard tolerance $n!\, \epsilon_{\text{mach}}\, s_{\max}(M_s)$, where $s_{\max}(M_s)$ is the largest singular value.

Table 4. Numerical rank verification for the orbit evaluation matrix.

| $n$ | $n!$ | scale $s$ | numerical rank | smallest singular value |
|---|---|---|---|---|
| 3 | 6 | 0.5 | 6 | $5.868 \times 10^{-3}$ |
| 3 | 6 | 1.0 | 6 | $5.030 \times 10^{-2}$ |
| 3 | 6 | 2.0 | 6 | $4.832 \times 10^{-1}$ |
| 4 | 24 | 0.5 | 24 | $1.181 \times 10^{-4}$ |
| 4 | 24 | 1.0 | 24 | $6.688 \times 10^{-3}$ |
| 4 | 24 | 2.0 | 24 | $1.145 \times 10^{-1}$ |
| 5 | 120 | 0.5 | 120 | $1.209 \times 10^{-8}$ |
| 5 | 120 | 1.0 | 120 | $2.688 \times 10^{-5}$ |
| 5 | 120 | 2.0 | 120 | $8.159 \times 10^{-2}$ |
| 6 | 720 | 0.5 | 720 | $2.619 \times 10^{-9}$ |
| 6 | 720 | 1.0 | 720 | $1.966 \times 10^{-5}$ |
| 6 | 720 | 2.0 | 720 | $4.555 \times 10^{-4}$ |

Table 4 shows full numerical rank in every tested case. The smallest singular value decreases for the most ill-conditioned setting $(n, s) = (6, 0.5)$, which is why we interpret these results only as finite-scale evidence rather than a proof of Conjecture 4.1. These experiments provide evidence for Conjecture 4.1, but do not constitute a proof for arbitrary $n$.

## C. Pairwise Insufficiency: A Gaussian Example

This appendix provides a concrete example supporting the gradient-level non-separability result in Appendix A and the discussion in Section 3.2. We construct two joint Gaussian distributions over $(z_1, z_2, z_3)$ that induce identical pairwise comparison probabilities $\mathbb{P}(z_i < z_j)$ for all $i \neq j$, while assigning different probabilities to the full-order event $\mathbb{P}(z_1 < z_2 < z_3)$.

Let $\mathbf{z} = (z_1, z_2, z_3) \sim \mathcal{N}(\mathbf{0}, \Sigma)$ with unit variances, i.e., $\mathrm{Var}(z_i) = 1$ for all $i$. For any $i \neq j$, the difference $z_i - z_j$ is a centered Gaussian random variable, so

$$\mathbb{P}(z_i < z_j) = \mathbb{P}(z_i - z_j < 0) = \frac{1}{2}, \qquad \forall\, i \neq j, \tag{19}$$

independently of the covariance structure $\Sigma$. Thus all such models share the same collection of pairwise comparison probabilities.

In contrast, the full-order probability depends on the joint distribution of consecutive differences. Define

$$u_1 = z_2 - z_1, \qquad u_2 = z_3 - z_2. \tag{20}$$

Then

$$\mathbb{P}(z_1 < z_2 < z_3) = \mathbb{P}(u_1 > 0,\ u_2 > 0), \tag{21}$$

and $(u_1, u_2)$ is a centered bivariate Gaussian random vector whose covariance depends on $\Sigma$. Writing $a = \mathrm{Cov}(z_1, z_2)$, $b = \mathrm{Cov}(z_2, z_3)$, and $c = \mathrm{Cov}(z_1, z_3)$, a short calculation gives

$$\rho = \mathrm{Corr}(u_1, u_2) = \frac{a + b - c - 1}{\sqrt{(2 - 2a)(2 - 2b)}}. \tag{22}$$

For a centered bivariate Gaussian with correlation $\rho$, the orthant probability admits the classical closed form

$$\mathbb{P}(u_1 > 0,\ u_2 > 0) = \frac{1}{4} + \frac{1}{2\pi} \arcsin(\rho). \tag{23}$$

Consider now the two covariance matrices

$$\Sigma^{(A)} = \begin{pmatrix} 1 & 0.6 & 0.2 \\ 0.6 & 1 & 0.6 \\ 0.2 & 0.6 & 1 \end{pmatrix}, \qquad \Sigma^{(B)} = \begin{pmatrix} 1 & 0.2 & 0.6 \\ 0.2 & 1 & 0.2 \\ 0.6 & 0.2 & 1 \end{pmatrix},$$

both of which are positive definite. In either case, all pairwise comparison probabilities equal $\frac{1}{2}$ as above. However, the corresponding correlations of $(u_1, u_2)$ differ: for $\Sigma^{(A)}$ we obtain $\rho^{(A)} = 0$ and hence $\mathbb{P}^{(A)}(z_1 < z_2 < z_3) = \frac{1}{4}$, while for $\Sigma^{(B)}$ we obtain $\rho^{(B)} = -0.75$ and thus

$$\mathbb{P}^{(B)}(z_1 < z_2 < z_3) = \frac{1}{4} + \frac{1}{2\pi} \arcsin(-0.75) \neq \frac{1}{4}.$$

This example shows that even within the restricted class of centered Gaussians with unit variances, identical pairwise comparison probabilities do not determine full-order probabilities. Pairwise statistics therefore provide a concrete instance of the more general gradient-level non-separability of surrogate objectives discussed in Section 3.2 and Appendix A.

## D. Symmetry of Ordering Events and Sequential Factorization Objectives

This appendix formalizes order-reversal symmetry at the level of ranking objectives and explains why Plackett–Luce-style sequential factorization objectives, such as ListNet (Cao et al., 2007) and ListMLE (Xia et al., 2008), violate this symmetry, whereas the proposed Full-Order Bound (FOB) preserves it.

Recall the order-reversal invariance notion from Section 3.3: a learning-to-rank objective $\mathcal{L}(\theta; \mathbf{x}, \mathbf{y}) = F(\mathbf{s}, \pi(\mathbf{y}))$ is *order-reversal invariant* if

$$F(\mathbf{s}, \pi(\mathbf{y})) = F(-\mathbf{s}, \pi(-\mathbf{y})) \qquad \text{for all } (\mathbf{s}, \mathbf{y}), \tag{24}$$

where $-\mathbf{y}$ is obtained by applying a strictly decreasing transformation to $\mathbf{y}$. Since $\mathbf{y}$ enters the objective only through the permutation it induces, the transformation $\mathbf{y} \mapsto -\mathbf{y}$ reverses that permutation, while $\mathbf{s} \mapsto -\mathbf{s}$ reverses the score ordering in the same way.

### D.1. Asymmetry of Plackett–Luce Objectives

We first show that Plackett–Luce-style objectives do not satisfy Equation (24). For concreteness, we work with the Plackett–Luce likelihood used in ListMLE; the same argument applies to related sequential factorization models.

Let $\mathbf{s} = (s_1, \ldots, s_n)$, and let $\pi$ be a permutation of $\{1, \ldots, n\}$ such that it orders the $y$'s as $y_{\pi_1} < \cdots < y_{\pi_n}$. The Plackett–Luce probability of $\pi$ is

$$\mathbb{P}_{\mathrm{PL}}(\pi \mid \mathbf{s}) = \prod_{k=1}^{n} \frac{\exp(s_{\pi_k})}{\sum_{j=k}^{n} \exp(s_{\pi_j})}, \tag{25}$$

and the corresponding (negative) log-likelihood loss is

$$\mathcal{L}_{\mathrm{PL}}(\theta; \mathbf{x}, \mathbf{y}) = F(\mathbf{s}, \pi(\mathbf{y})) = -\log \mathbb{P}_{\mathrm{PL}}(\pi_{\mathbf{y}} \mid \mathbf{s}), \tag{26}$$

where $\pi_{\mathbf{y}}$ is the permutation induced by $\mathbf{y}$ (labels sorted from least to most relevant). Under the order-reversal transformation $(\mathbf{s}, \mathbf{y}) \mapsto (-\mathbf{s}, -\mathbf{y})$, the induced permutation becomes the reversed permutation $\pi_{-\mathbf{y}}$, so the transformed loss is

$$F(-\mathbf{s}, \pi(-\mathbf{y})) = -\log \mathbb{P}_{\mathrm{PL}}(\pi_{-\mathbf{y}} \mid -\mathbf{s}). \tag{27}$$

We now show that, for $n \geq 3$, these two losses differ in general. Consider $n = 3$ with labels indexed such that $y_1 < y_2 < y_3$, so that $\pi_{\mathbf{y}} = (1, 2, 3)$ and $\pi_{-\mathbf{y}} = (3, 2, 1)$. Write

$$a = \exp(s_1), \quad b = \exp(s_2), \quad c = \exp(s_3),$$

so that $a, b, c > 0$. Then

$$\mathbb{P}_{\mathrm{PL}}(\pi_{\mathbf{y}} \mid \mathbf{s}) = \frac{a}{a+b+c} \cdot \frac{b}{b+c} = \frac{ab}{(a+b+c)(b+c)}, \tag{28}$$

$$\mathbb{P}_{\mathrm{PL}}(\pi_{-\mathbf{y}} \mid -\mathbf{s}) = \frac{1/c}{1/a + 1/b + 1/c} \cdot \frac{1/b}{1/a + 1/b} = \frac{ab}{ab + ac + bc} \cdot \frac{a}{a+b}. \tag{29}$$

Equality of these two expressions for all $(a, b, c)$ would require

$$\frac{ab}{(a+b+c)(b+c)} = \frac{ab}{ab+ac+bc} \cdot \frac{a}{a+b},$$

which simplifies to

$$b^2 = ac.$$

Thus $F(\mathbf{s}, \pi(\mathbf{y})) = F(-\mathbf{s}, \pi(-\mathbf{y}))$ only on the measure-zero set of scores satisfying $b^2 = ac$ (equivalently, $2s_2 = s_1 + s_3$), and in general

$$F(\mathbf{s}, \pi(\mathbf{y})) \neq F(-\mathbf{s}, \pi(-\mathbf{y})).$$

For general $n \geq 3$, embedding this configuration into the first three positions (with the remaining items assigned scores far from $s_1, s_2, s_3$ so that their selection factors are essentially unaffected) yields the same strict inequality, so order-reversal invariance fails for every $n \geq 3$.

Therefore, Plackett–Luce and ListMLE-style objectives are not order-reversal invariant in the sense of Equation (24). The sequential factorization introduces an intrinsic directionality: the likelihood of a given ordering depends not only on the relative arrangement of items but also on the stage at which each item is selected.

## D.2. Order-Reversal Symmetry of the Full-Order Bound

*Proof of Proposition 4.2. Setup.* Recall the latent location–scale model $z_i = \mu_i + \sigma_i \varepsilon_i$ with i.i.d. noise $\varepsilon_i \sim p_0$ and a symmetric base density, $p_0(\varepsilon) = p_0(-\varepsilon)$ (e.g., Gaussian, logistic, Laplace). For a strictly ordered instance we reindex so that $\pi(\mathbf{y}) = (1, \ldots, n)$ and the observed event is $\mathcal{E} = \{z_1 \leq \cdots \leq z_n\}$. For ordered cut points $\mathbf{t} \in \mathcal{T} = \{\mathbf{t} : t_1 \leq \cdots \leq t_{n-1}\}$ with $t_0 = -\infty$, $t_n = +\infty$, the bound is

$$\text{FOB}(\mathbf{t}) = \sum_{i=1}^{n} \log \mathbb{P}(t_{i-1} \leq z_i \leq t_i),$$

and the order-level objective is $F_{\text{FOB}}(\boldsymbol{\mu}, \boldsymbol{\sigma}; \pi(\mathbf{y})) = -\max_{\mathbf{t} \in \mathcal{T}} \text{FOB}(\mathbf{t})$, a deterministic functional of the predicted parameters $(\boldsymbol{\mu}, \boldsymbol{\sigma})$.

*Reversed instance.* The order-reversal operator $\mathcal{R} : (\mu_i, \sigma_i) \mapsto (\mu'_i, \sigma'_i)$ with

$$\mu'_i = -\mu_{n+1-i}, \qquad \sigma'_i = \sigma_{n+1-i}, \qquad i = 1, \ldots, n,$$

negates the predicted locations and carries the dispersions along by the same reindexing; together with the label reversal $\pi(\mathbf{y}) \mapsto \pi(-\mathbf{y})$ this is the transformation of Definition 3.3 for the latent model. Equivalently the reversed latent scores are $z'_i = -z_{n+1-i}$, whose observed event is again $\{z'_1 \leq \cdots \leq z'_n\}$. Let $\text{FOB}'$ denote the bound of this reversed instance, so that $F_{\text{FOB}}(\mathcal{R}(\boldsymbol{\mu}, \boldsymbol{\sigma}); \pi(-\mathbf{y})) = -\max_{\mathbf{u} \in \mathcal{T}} \text{FOB}'(\mathbf{u})$.

*Step 1 (cut-point bijection).* Define $\Phi(\mathbf{t}) = \mathbf{t}'$ by $t'_i = -t_{n-i}$ for $i = 1, \ldots, n-1$ (consistently $t'_0 = -t_n = -\infty$ and $t'_n = -t_0 = +\infty$). If $t_1 \leq \cdots \leq t_{n-1}$ then $-t_{n-1} \leq \cdots \leq -t_1$, i.e. $t'_1 \leq \cdots \leq t'_{n-1}$, so $\Phi$ maps $\mathcal{T}$ into $\mathcal{T}$; since $\Phi$ is its own inverse, it is a bijection of $\mathcal{T}$.

*Step 2 (per-interval invariance).* Fix $\mathbf{t} \in \mathcal{T}$ and let $\mathbf{t}' = \Phi(\mathbf{t})$. The interval flip $\mathbb{P}(t_{i-1} \leq z_i \leq t_i) = \mathbb{P}(-t_i \leq -z_i \leq -t_{i-1})$ holds for any random variable. By symmetry of $p_0$, $-z_i = -\mu_i + \sigma_i(-\varepsilon_i)$ has the same law as the reversed model's $z'_{n+1-i}$ (location $-\mu_i$, scale $\sigma_i$); moreover $-t_i = t'_{n-i}$ and $-t_{i-1} = t'_{n+1-i}$. Hence

$$\mathbb{P}(t_{i-1} \leq z_i \leq t_i) = \mathbb{P}\big(t'_{n-i} \leq z'_{n+1-i} \leq t'_{n+1-i}\big),$$

i.e., the $i$-th interval probability of the original instance equals the $(n+1-i)$-th interval probability of the reversed instance.

*Step 3 (equality of maxima).* Summing the logarithms over $i$ and reindexing $j = n+1-i$,

$$\text{FOB}(\mathbf{t}) = \sum_{i=1}^{n} \log \mathbb{P}(t_{i-1} \leq z_i \leq t_i) = \sum_{j=1}^{n} \log \mathbb{P}\big(t'_{j-1} \leq z'_j \leq t'_j\big) = \text{FOB}'(\Phi(\mathbf{t})).$$

Because $\Phi$ is a bijection of $\mathcal{T}$,

$$\max_{\mathbf{t} \in \mathcal{T}} \text{FOB}(\mathbf{t}) = \max_{\mathbf{t} \in \mathcal{T}} \text{FOB}'(\Phi(\mathbf{t})) = \max_{\mathbf{u} \in \mathcal{T}} \text{FOB}'(\mathbf{u}),$$

and therefore $F_{\text{FOB}}(\boldsymbol{\mu}, \boldsymbol{\sigma}; \pi(\mathbf{y})) = -\max_{\mathbf{t}} \text{FOB}(\mathbf{t}) = -\max_{\mathbf{u}} \text{FOB}'(\mathbf{u}) = F_{\text{FOB}}(\mathcal{R}(\boldsymbol{\mu}, \boldsymbol{\sigma}); \pi(-\mathbf{y}))$, establishing order-reversal invariance in the sense of Definition 3.3. $\square$

# E. Convexity of the Inner Optimization

This appendix provides the proof of Theorem 5.1.

Fix latent score parameters (implicitly absorbed into the densities below), and assume that $z_1, \ldots, z_n$ are independent with (Lebesgue) densities $p_i$ on $\mathbb{R}$. We assume each $p_i$ is *log-concave*, i.e.,

$$p_i(\lambda x + (1 - \lambda) y) \geq p_i(x)^\lambda p_i(y)^{1-\lambda}, \qquad \forall x, y \in \mathbb{R}, \ \lambda \in [0, 1].$$

Recall $t_0 = -\infty$, $t_n = +\infty$, and the feasible set $\mathcal{T} = \{\mathbf{t} : t_1 \leq \cdots \leq t_{n-1}\}$. For $\mathbf{t} \in \mathcal{T}$, the Full-Order Bound is

$$\text{FOB}(\mathbf{t}) = \sum_{i=1}^{n} \log \mathbb{P}\big(t_{i-1} \leq z_i \leq t_i\big) = \sum_{i=1}^{n} \log \int_{t_{i-1}}^{t_i} p_i(x) \, dx. \tag{30}$$

We first show that for a log-concave density, the probability mass of an interval is log-concave in the endpoints.

**Lemma E.1** (Log-concavity of interval integrals). *Let $p$ be a log-concave density on $\mathbb{R}$ and define*

$$F(a, b) \triangleq \int_a^b p(x) \, dx, \qquad a \le b.$$

*Then $F(a, b)$ is log-concave on the domain $\{(a, b) : a \le b\}$. Equivalently, $g(a, b) \triangleq \log F(a, b)$ is jointly concave on $\{a \le b\}$.*

*Proof.* Define the indicator of the interval as a function of $(a, b, x)$:

$$\mathbb{I}_{(a,b)}(x) = \mathbb{I}\{a \le x \le b\}.$$

Consider the function

$$f(a, b, x) \triangleq p(x) \, \mathbb{I}\{a \le x \le b\}, \qquad (a, b, x) \in \mathbb{R}^3.$$

We claim $f$ is log-concave on its support $\{(a, b, x) : a \le x \le b\}$. Indeed, $p(x)$ is log-concave in $x$ by assumption, and the set $\{(a, b, x) : a \le x \le b\}$ is convex in $(a, b, x)$. Moreover, the indicator of a convex set is log-concave in the extended sense (its logarithm is $0$ on the set and $-\infty$ outside), hence the product of log-concave functions is log-concave on $\mathbb{R}^3$.

Now observe that

$$F(a, b) = \int_{\mathbb{R}} f(a, b, x) \, dx.$$

A standard consequence of the Prékopa–Leindler theorem states that the marginal of a log-concave function is log-concave: if $f(u, x)$ is log-concave in $(u, x)$, then $u \mapsto \int f(u, x) \, dx$ is log-concave. Applying this to $u = (a, b)$ yields that $(a, b) \mapsto F(a, b)$ is log-concave on $\{a \le b\}$, and thus $g(a, b) = \log F(a, b)$ is jointly concave on $\{a \le b\}$. □

**Proof of Theorem 5.1.** For each $i$, define

$$g_i(t_{i-1}, t_i) \triangleq \log \int_{t_{i-1}}^{t_i} p_i(x) \, dx, \qquad t_{i-1} \le t_i.$$

By Lemma E.1, $g_i$ is jointly concave in $(t_{i-1}, t_i)$. Using (30), we can write

$$\text{FOB}(\mathbf{t}) = \sum_{i=1}^n g_i(t_{i-1}, t_i).$$

A sum of concave functions is concave, hence $\text{FOB}(\mathbf{t})$ is concave on the domain where all intervals are valid; restricting to $\mathcal{T}$ (which is a convex polyhedral cone defined by linear inequalities) preserves concavity. Therefore, the inner problem

$$\max_{\mathbf{t} \in \mathcal{T}} \text{FOB}(\mathbf{t})$$

is a convex optimization problem (maximization of a concave objective over a convex feasible set), completing the proof. □

## F. Tightening with Tied Labels

In many ranking benchmarks, labels take values in a finite set $y_i \in \{1, \ldots, k\}$, and multiple items may share the same label. In this case, we use label-level cut points $\mathbf{t} = (t_1, \ldots, t_{k-1})$ to define label intervals $(t_{j-1}, t_j)$, with $t_0 = -\infty$ and $t_k = +\infty$. When ties are abundant, a direct per-item tightening objective of the form $\sum_i \log \mathbb{P}(t_{y_i-1} \le z_i \le t_{y_i})$ can be overly sensitive to a few extreme items within a label bin and can also amplify the influence of bins that contain many tied items, which empirically leads to unstable cut-point updates.

To improve robustness under ties and reduce label-bin imbalance, we tighten cut points using an aggregated objective that first sums interval probabilities within each label bin and then applies a logarithm:

$$\text{FOB}_{\text{agg}}(\mathbf{t}) = \sum_{j=1}^k \log \left( \sum_{i \in \mathcal{I}_j} \mathbb{P}(t_{j-1} \le z_i \le t_j) \right), \tag{31}$$

where $\mathcal{I}_j = \{i : y_i = j\}$ and $\mathbf{t} \in \mathcal{T} = \{\mathbf{t} : t_1 \leq \cdots \leq t_{k-1}\}$. This aggregation has two practical effects: (i) it compresses the scale disparity across label bins, since each bin contributes a single log term rather than a sum of $|\mathcal{I}_j|$ per-item log terms whose magnitude grows with the bin size, and (ii) it reduces the impact of individual outliers by allowing items within the same label bin to share a normalized "responsibility" in the gradient of the bin term.

For log-concave base densities, each individual interval probability $(a, b) \mapsto \mathbb{P}(a \leq z_i \leq b)$ is log-concave in $(a, b)$ by Lemma E.1. However, Equation (31) involves a logarithm of a *sum* of such terms, and log-concavity is not preserved under summation in general. Therefore, unlike the strict-order per-item objective in Section 5.2, $\text{FOB}_{\text{agg}}$ does not admit a global concavity guarantee over $\mathcal{T}$ without additional assumptions. In our experiments, however, optimizing $\text{FOB}_{\text{agg}}$ with the same safe-region updates as in Section 5.3 acts as a stable constrained-ascent heuristic and consistently improves cut-point stability under tied labels.

After obtaining the approximate $\mathbf{t}^*$ via the same safe-region gradient ascent (SRGA) procedure based on $\text{FOB}_{\text{agg}}(\mathbf{t})$, we can use the original per-item bound $\text{FOB}(\mathbf{t}^*; \theta)$ to update the model parameter $\theta$.

## G. LAMBDAFOB: A Metric-Aligned Extension of FOB

This appendix describes LAMBDAFOB, a metric-aligned extension of FOB used only in experiments to enable a fair comparison with Lambda-style baselines.

We use an exponential-discount formulation of DCG. Let $g_i = 2^{y_i} - 1$ be the relevance gain of item $i$ and $\gamma \in (0, 1)$ a decay factor. For a ranked list, let $r_i \in \{0, 1, 2, \dots\}$ denote the rank position of item $i$, where ranks start with 0 for the top-ranked item. The DCG can be written as a sum over items:

$$\text{DCG} = \sum_{i=1}^{n} g_i \, \gamma^{r_i}.$$

Given cut points $\mathbf{t}^*$, each item induces a soft distribution over intervals

$$p_{i,k}(\mathbf{t}^*; \theta) \triangleq \mathbb{P}_\theta\left(t_{k-1}^* \leq z_i \leq t_k^*\right), \qquad k = 1, \dots, K. \tag{32}$$

Let $c_k = \sum_{i=1}^{n} p_{i,k}$ denote the expected number of valid items in interval $k$. The starting rank offset of interval $k$ in the final list is defined as

$$o_k \triangleq \sum_{j > k} c_j, \tag{33}$$

so that items in higher-score intervals occupy earlier positions. We associate interval $k$ with the discount factor

$$D_k \triangleq \gamma^{o_k}.$$

To construct a listwise metric surrogate, we consider inserting item $i$ into interval $k$. The direct gain of this insertion is $g_i D_k$. Inserting into interval $k$ shifts all items currently assigned to intervals $1, 2, \dots, k$ one position later in the ranked list, which multiplies their discount by $\gamma$. The induced loss therefore equals $(1 - \gamma)$ times the discounted DCG contributed by those items, excluding $i$ itself. We define

$$S_{\neg i}(k) \triangleq \sum_{j \neq i} \sum_{l \leq k} p_{j,l} \, g_j \, D_l. \tag{34}$$

The probabilistic insertion value is then

$$V_{i,k} \triangleq g_i \, D_k - (1 - \gamma) \, S_{\neg i}(k).$$

Here $V_{i,k}$ represents the incremental contribution to exponentially-discounted DCG (and hence NDCG) obtained by inserting item $i$ into interval $k$: the first term accounts for the direct gain of placing $i$ at the corresponding rank offset, while the second term subtracts the discount shrinkage induced on existing items other than item $i$.

Let the current interval of item $i$ (determined by $(\boldsymbol{\mu}, \mathbf{t}^*)$) define a baseline value, which is treated as a constant (stop-gradient). The expected relative gain of item $i$ is

$$\Delta_i(\mathbf{t}^*; \theta) = \sum_{k=1}^{K} p_{i,k}(\mathbf{t}^*; \theta)(V_{i,k} - \text{baseline}_i). \tag{35}$$

The final per-query LAMBDAFOB training loss is

$$\mathcal{L}_{\text{LAMBDAFOB}} = -\text{FOB}(\mathbf{t}^*; \theta) - \alpha \sum_{i=1}^{n} \widetilde{\Delta}_i(\mathbf{t}^*; \theta), \tag{36}$$

where $\alpha = 0.5$ in all experiments and $\widetilde{\Delta}_i$ denotes per-query normalization with stop-gradient statistics. The auxiliary term provides a metric-aligned shaping signal while preserving the original inner tightening structure of FOB.

## H. Experimental Details

*Table 5.* Training configurations for the MNIST and Web30K/Istella experiments.

| | **MNIST** | **Web30K / Istella** |
|---|---|---|
| **Backbone** | | |
| Architecture | CNN + MLP (LeCun et al., 2002) | 3-layer MLP |
| Activations | ReLU (Glorot et al., 2011) | ReLU |
| Normalization in backbone | – | BatchNorm (Ioffe & Szegedy, 2015) |
| **FOB-related parameters** | | |
| Latent distribution | Heteroscedastic Normal | Homoscedastic Logistic |
| List-wise normalization | Batch-wise | List-wise |
| Cut-point initialization | Quantiles of $\{\mu_i\}$ | Quantiles of standard logistic |
| List-length warmup | Yes | No |
| Main objective | $\max_\theta \ \text{FOB}(\mathbf{t}^*; \theta)$ | $\max_\theta \ \text{FOB}(\mathbf{t}^*; \theta)$ |
| **SRGA-related parameters** | | |
| Inner objective | $\mathbf{t}^* = \arg\max_{\mathbf{t}} \ \text{FOB}(\mathbf{t})$ | $\mathbf{t}^* = \arg\max_{\mathbf{t}} \ \text{FOB}_{\text{agg}}(\mathbf{t})$ |
| SRGA steps | 40 | 10 |
| SRGA learning rate | 0.01 | 0.4 |
| SRGA decay rate | 0.98 | 0.8 |
| **Optimization / training parameters** | | |
| Optimizer | Adam (Kingma & Ba, 2015) | Adam |
| Batch size | 100 | 32 |
| Training budget | 100,000 ($n \leq 8$), 200,000 ($n > 8$) | 1 epoch |
| Initial learning rate | $10^{-3}$ | $10^{-3.5}$ |

This appendix summarizes the experimental configurations used in the MNIST and Web30K/Istella benchmarks. The two settings differ substantially in label structure, list length, and data scale, and therefore employ different distributional assumptions and tightening objectives. Key hyperparameters are reported in Table 5.

For the MNIST experiments, we assume a heteroscedastic normal latent distribution, with item-specific variances predicted by the model. The inner optimization maximizes the standard Full-Order Bound $\text{FOB}(\mathbf{t})$ using safe-region gradient ascent (SRGA). Cut points are initialized using empirical quantiles of the predicted means $\{\mu_i\}$ within each list, and latent variables are normalized within each mini-batch before tightening. When training LambdaRank baselines on MNIST, we compute $\lambda$ using classification accuracy as the target metric.

For the Web30K and Istella benchmarks, we assume a homoscedastic logistic latent distribution with a fixed scale parameter. Due to the presence of tied relevance labels, the inner optimization maximizes the aggregated objective $\text{FOB}_{\text{agg}}(\mathbf{t})$ defined in Appendix F. Cut points are initialized using quantiles of the standard logistic distribution, and list-wise normalization is applied prior to tightening. For LambdaRank baselines in this setting, $\lambda$ is computed using full-list NDCG as the optimization target.

## I. Ablation on Tightening and Safe Region

We investigate the effect of the proposed tightening procedure through controlled ablation studies on the MNIST permutation task with $n = 8$, conducting five runs with different random seeds. Unless otherwise noted, we report the mean performance $\pm$ the radius of the 95% confidence interval, where the radius is computed as approximately 2.78 times the standard error.

*Table 6.* Ablation results on MNIST permutation task ($n = 8$).

| Method Variant | ACC (%) |
|---|---|
| Full FOB | $68.20 \pm 1.61$ |
| w/o warmup (fixed list length 8) | $63.71 \pm 7.46$ |
| w/o normalization (Equation (9)) | $50.26 \pm 33.36$ |
| w/o safe region (re-sort after update) | $62.12 \pm 17.67$ |
| w/o tightening (fixed cut points) | $48.60 \pm 16.23$ |

Removing any of the key components leads to a pronounced drop in stability, as reflected by the much larger confidence intervals across runs. Eliminating the list-wise normalization in Equations (8) and (9) causes the model to spend capacity on globally shifting and rescaling the latent scores, directions to which the FOB objective is invariant. As a result, training frequently diverges or converges to degenerate solutions, yielding both low accuracy and very wide confidence intervals.

Fixing the cut points instead of tightening them ("w/o tightening") effectively decouples the objective from the true full-order probability and turns FOB into a crude surrogate. This significantly degrades final performance and again increases the uncertainty, highlighting the importance of aligning the training signal with the underlying full-order objective via inner maximization over $\mathbf{t}$.

On the optimization side, removing the safe-region mechanism ("w/o safe region") and simply re-sorting the cut points after each gradient step leads to unstable inner optimization: the cut-point updates tend to zigzag along the boundary of the feasible region, mirroring the behavior of aggressive projected methods discussed in Section 5.3, and in turn induce noisy outer updates on $\theta$. Removing the list-length warmup increases the sharpness of the tightened bound $\max_{\mathbf{t}} \text{FOB}(\mathbf{t}; \theta)$ from the very beginning of training. In this case, early gradient steps have an outsized influence on the eventual convergence point and amplify sensitivity to initialization and learning rate.

Together, these ablations confirm that normalization, tightening, safe-region updates, and warmup contribute jointly to the stability and effectiveness of the FOB-based training procedure.

## J. Effect of Latent Score Distributions

We next investigate the robustness of the proposed method to the choice of latent score distribution on the same MNIST permutation task with $n = 8$. We use the same confidence-interval convention as in Appendix I.

*Table 7.* Effect of latent score distributions on MNIST permutation task ($n = 8$).

| Latent Distribution | ACC (%) |
|---|---|
| Gaussian | $68.20 \pm 1.61$ |
| Logistic | $58.28 \pm 18.98$ |
| Laplace | $64.54 \pm 5.48$ |
| Cauchy | $62.05 \pm 3.31$ |

Overall, the method attains reasonable accuracy under several choices of location–scale family, but both the best performance and the tightest confidence intervals are obtained with the Gaussian latent model. Replacing the Gaussian with a logistic distribution, which is still log-concave, leads to noticeably higher variability across runs and reduced average accuracy, indicating that the optimization becomes less stable even though the convexity argument in Appendix E still applies. The Laplace latent model, also log-concave, performs slightly worse than Gaussian but remains comparatively stable.

In contrast, using a Cauchy latent distribution yields lower peak performance. Unlike the previous families, Cauchy is not log-concave, so the interval log-probability no longer enjoys the concavity guarantee underlying Theorem 5.1. The resulting

loss landscape is less well behaved, and the optimizer struggles to match the accuracy achieved by log-concave latent models. These results support our choice of Gaussian latent scores as a default: it offers a favorable combination of theoretical convexity properties for the inner problem and empirical robustness in terms of both accuracy and training stability.

## K. Effect of SRGA Steps

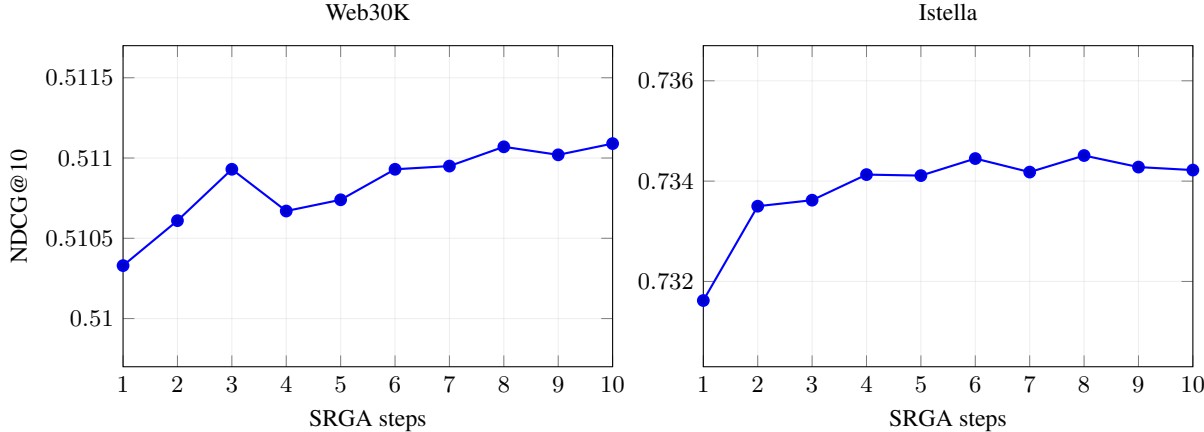

*Figure 2.* Effect of the number of SRGA tightening steps on NDCG@10.

Figure 2 examines the effect of the number of inner SRGA iterations used to tighten the cut points in FOB. On both Web30K and Istella, increasing the number of SRGA steps leads to clear performance gains in the early regime (the first three to four steps), indicating that even limited tightening substantially improves the quality of the lower bound and strengthens the resulting outer training signal. Beyond this point, the curves quickly plateau, suggesting that the inner optimization is already close to convergence after only a few iterations. This empirical behavior supports the use of a small SRGA budget in practice: a moderate number of inner steps (e.g., 5–10 for Web30K and Istella) captures most of the achievable improvement while keeping the additional computational overhead well controlled.

## L. Effect of Order Reversal

*Table 8.* Sensitivity to label order reversal for ListMLE and the proposed FOB.

| Method | Web30K | | | Istella | | |
|---|---|---|---|---|---|---|
| | N@10 | N@20 | Kendall's $\tau$ | N@10 | N@20 | Kendall's $\tau$ |
| ListMLE | $49.66 \pm 0.13$ | $53.81 \pm 0.13$ | $33.66 \pm 0.21$ | $71.12 \pm 0.42$ | $77.83 \pm 0.36$ | $83.90 \pm 0.35$ |
| ListMLE (rev) | $50.76 \pm 0.25$ | $54.92 \pm 0.20$ | $33.94 \pm 0.27$ | $73.29 \pm 0.30$ | $79.86 \pm 0.25$ | $85.67 \pm 0.10$ |
| FOB | $51.11 \pm 0.16$ | $55.17 \pm 0.11$ | $34.05 \pm 0.22$ | $73.44 \pm 0.22$ | $79.99 \pm 0.18$ | $86.21 \pm 0.08$ |
| FOB (rev) | $51.07 \pm 0.17$ | $55.16 \pm 0.13$ | $33.99 \pm 0.24$ | $73.42 \pm 0.21$ | $79.98 \pm 0.17$ | $86.21 \pm 0.08$ |

This appendix provides an empirical validation of the order-reversal invariance property discussed in Section 3.3 and Appendix D. While the discussions above establish this property theoretically, here we examine its practical implications by explicitly reversing label orders during training.

For each dataset (Web30K and Istella), we construct a reversed-label version of the training data by applying a strictly decreasing transformation to the relevance labels, which induces the reversed permutation for every query. Models are trained on the reversed labels using the same architectures, hyperparameters, and optimization procedures as in the original setting. At evaluation time, predictions and labels are reversed back to the original order so that all methods are compared under identical metrics.

Table 8 reports ranking performance for ListMLE and FOB under the original labels and their reversed counterparts. For ListMLE, reversing the label order leads to substantial changes in performance across all metrics and both datasets; in fact,

the reversed-label models perform even better than in the original setting. In contrast, FOB exhibits nearly identical results before and after label reversal, with differences well within statistical variation.

These results highlight a concrete consequence of order-reversal asymmetry in sequential factorization models. ListMLE defines ordering probabilities through a directional Plackett–Luce factorization, and reversing the label order effectively changes the learning objective, yielding a different optimum in parameter space. As a result, training on reversed labels is not equivalent to training on the original labels. Crucially, which of the two directions performs better is dataset-dependent and cannot be known a priori, so the asymmetry is a liability regardless of its sign: it forces an arbitrary modeling choice that materially affects the learned ranker. FOB removes this choice entirely.

By contrast, FOB is constructed directly from the ordering event itself and preserves order-reversal invariance by design. Reversing labels corresponds to an equivalent reparameterization of the latent ordering event, leaving the tightened full-order bound unchanged. The empirical insensitivity of FOB to label reversal therefore provides practical confirmation of the theoretical symmetry analysis in Section 3.3 and Appendix D.

# M. Computational Efficiency

We next compare the computational cost of FOB-based training against representative learning-to-rank baselines. We report the time per mini-batch (in milliseconds) for RankNet, ListMLE, NeuralSort, DiffSort, and FOB with Safe-Region Gradient Ascent (SRGA) using $K \in \{40, 80\}$ inner steps. Results as a function of list length are summarized in Table 9. We omit other pairwise/listwise surrogates and EF-DSF for clarity, since their empirical timing behavior is similar to the methods already shown here.

*Table 9.* Training time in the MNIST task per mini-batch (ms) as a function of list length. FOB uses Safe-Region Gradient Ascent (SRGA) with $K \in \{40, 80\}$ inner steps.

| Method | $n = 8$ | $n = 16$ | $n = 32$ | $n = 64$ | $n = 128$ |
|---|---|---|---|---|---|
| RankNet | 30 | 35 | 59 | 97 | 189 |
| ListMLE | 28 | 37 | 59 | 99 | 187 |
| NeuralSort | 29 | 38 | 60 | 98 | 187 |
| DiffSort (bitonic) | 38 | 51 | 86 | 143 | 246 |
| FOB ($K = 40$) | 31 | 39 | 64 | 102 | 194 |
| FOB ($K = 80$) | 34 | 43 | 67 | 104 | 198 |

As shown in Table 9, RankNet, ListMLE, NeuralSort, and FOB all have very similar practical cost in our implementation, whereas DiffSort is consistently more expensive across all tested list lengths. In particular, the timing gap among RankNet, ListMLE, NeuralSort, and FOB remains small throughout the tested range, indicating that the SRGA-based inner tightening does not introduce prohibitive overhead in practice.

Another notable observation is that increasing the number of SRGA inner steps from $K = 40$ to $K = 80$ introduces only a very small additional wall-clock overhead. This is consistent with the implementation of SRGA as a lightweight inner update over the auxiliary cut points while keeping the model outputs fixed, and further highlights the practical efficiency of the tightening procedure.

Compared with differentiable-sorting baselines such as DiffSort, FOB is consistently more efficient across all tested list lengths, while remaining competitive with standard pairwise and sequential baselines. Overall, these results show that directly optimizing a tightened full-order objective does not impose prohibitive computational cost: with an efficient implementation, FOB achieves training efficiency close to simpler ranking losses and substantially better efficiency than DiffSort.

Finally, these differences affect only the training phase. At inference time, all methods use the same deterministic scoring model $f_\theta$ and produce rankings by sorting scores (e.g., the means $\mu_i$), so their per-query inference cost is essentially identical. The SRGA inner optimization is only performed during training.

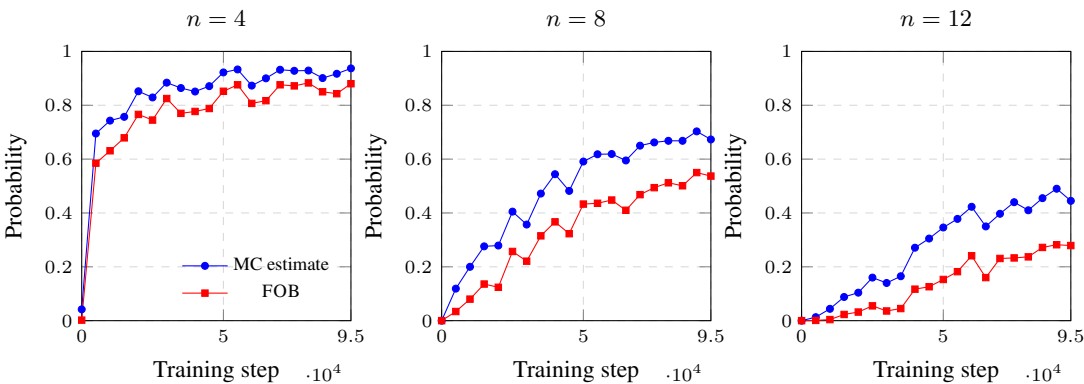

*Figure 3.* Empirical calibration of FOB against the full-order probability for list sizes $n = 4, 8, 12$. In each panel, the blue curve shows the Monte Carlo estimate of the exact full-order probability $\mathbb{P}(z_1 \leq \cdots \leq z_n)$, and the red curve shows the optimized FOB lower bound during training. The lower bound consistently improves together with the true probability across all list sizes. As $n$ increases, the gap becomes larger, but FOB still tracks the optimization trend of the exact full-order probability and remains informative during training.

## N. Empirical Tightness of the Full-Order Bound

To further assess the tightness of FOB, we conducted additional calibration experiments on list sizes $n \in \{4, 8, 12\}$, where the exact full-order probability can still be accurately approximated by Monte Carlo (MC) sampling. Specifically, during training, we periodically estimated the target probability

$$\mathbb{P}(z_1 \leq \cdots \leq z_n)$$

using MC samples, and compared it against the optimized FOB lower bound at the same training step.

Figure 3 shows that across all three list sizes, the optimized FOB lower bound consistently tracks the true full-order probability throughout training. As expected, the bound is relatively loose at initialization, since the model is untrained and the optimized cut points are not yet aligned with the target ordering event. As training proceeds, both the MC-estimated probability and the FOB lower bound increase substantially, and the bound becomes progressively tighter.

The calibration is strongest for the smallest list size $n = 4$, where the lower bound closely follows the true full-order probability throughout training. As the list size increases to $n = 8, 12$, the gap between the MC estimate and the FOB lower bound becomes larger, which is expected since the target event becomes increasingly constrained. Nevertheless, even for $n = 12$, the bound still improves monotonically with training and tracks the same optimization trend as the true full-order probability.

These results provide empirical support that FOB is not merely a valid lower bound in theory, but also remains practically informative during optimization across a range of list sizes. In particular, while the bound becomes looser for larger $n$, it still captures the training progress of the exact full-order probability and provides a meaningful non-vanishing learning signal. This complements our main-paper ablations, which already show that the tightening step is important for both accuracy and optimization stability.

## O. Sensitivity to SRGA hyperparameters

We further examine the sensitivity of the SRGA inner-loop hyperparameters on the MNIST permutation task with list size $n = 4$. We compare five settings ranging from fine to coarse optimization budgets: the finer settings use more SRGA steps together with a smaller learning rate and a slower decay schedule, while the coarser settings use fewer steps with a larger initial learning rate and a faster decay.

Table 10 shows that FOB is fairly stable across a wide range of SRGA hyperparameters. The final ranking accuracy varies only mildly, from $88.91 \pm 0.83$ to $89.88 \pm 0.78$, without any abrupt degradation. As expected, finer optimization settings consistently lead to slightly better final performance, suggesting that a more accurate solution of the inner tightening problem can translate into modest gains. At the same time, even substantially cheaper settings remain competitive, indicating that the method is not overly sensitive to SRGA hyperparameter tuning.

*Table 10.* Sensitivity of FOB to SRGA hyperparameters on the MNIST permutation task with $n = 4$.

| SRGA steps $K$ | 40 | 30 | 20 | 15 | 10 |
|---|---|---|---|---|---|
| Learning rate | 0.01 | 0.02 | 0.05 | 0.07 | 0.10 |
| Decay ratio | 0.98 | 0.95 | 0.90 | 0.85 | 0.80 |
| ACC (%) | $89.88 \pm 0.78$ | $89.48 \pm 0.23$ | $89.26 \pm 0.67$ | $89.24 \pm 0.66$ | $88.91 \pm 0.83$ |

This behavior is consistent with the optimization structure of FOB: the SRGA inner problem is convex (equivalently, maximizing a concave objective over the feasible region), so the landscape is well-behaved and does not suffer from spurious local optima. Consequently, using a finer optimization budget tends to improve convergence quality, while relatively coarse settings still produce stable results.

## P. Role of $\sigma$

To examine whether the variance parameter $\sigma$ provides meaningful modeling capacity beyond global rescaling, we compare a heteroscedastic variant, where each item predicts its own uncertainty, against a homoscedastic variant, where all items share the same variance. Table 11 reports the results on the MNIST permutation task with $n = 4$.

*Table 11.* Effect of heteroscedastic vs. homoscedastic variance modeling on the MNIST permutation task ($n = 4$).

| Type | ACC (%) |
|---|---|
| Heteroscedastic | $89.88 \pm 0.78$ |
| Homoscedastic | $88.48 \pm 0.77$ |

The heteroscedastic variant outperforms the homoscedastic one, indicating that the variance parameters are not redundant degrees of freedom. Intuitively, some MNIST digit compositions are more ambiguous than others. The heteroscedastic formulation allows the latent model to adapt its spread to item-dependent ambiguity, assigning greater uncertainty to harder examples while keeping easier examples sharper. This suggests that $\sigma$ plays a substantive role in modeling uncertainty, rather than merely introducing an additional but unnecessary scaling parameter.

## Q. Results in the Long-List Scenario

To examine the long-list setting more directly, we conducted an additional experiment with very large list sizes, specifically $n = 256$ and $n = 512$, and measured performance using Kendall's $\tau$. Here each composite image encodes a 5-digit number rather than the 4-digit number used in Section 6.1, since the larger label range keeps items distinct and reduces ties when the list contains up to 512 items. The results are reported in Table 12.

*Table 12.* Kendall's $\tau$ (%) in the long-list scenario. Results are from a single run per setting and should be interpreted as an indicative stress test.

| Method | $n = 256$ | $n = 512$ |
|---|---|---|
| RankNet | 86.06 | 87.29 |
| ListMLE | 85.39 | 86.72 |
| DiffSort | 67.80 | 63.75 |
| FOB | 96.77 | 96.06 |

As shown in Table 12, FOB maintains a clear advantage over all baselines even when the list length increases to 256 and 512. In particular, the margin over RankNet and ListMLE remains substantial, while the gap over DiffSort becomes even more pronounced. These results provide preliminary evidence that FOB continues to yield a meaningful and effective learning signal in the very long-list regime, where surrogate or differentiable-sorting approaches may become increasingly misaligned with the underlying full-order structure.

