# OpenReview forum: "Learning to Rank by Directly Optimizing Full-Order Probabilities"
_ICML.cc/2026/Conference — ICML 2026 regular_

### Official Review · Reviewer_a3j8 · 2026-02-26

**Soundness:** 2
**Presentation:** 2
**Significance:** 3
**Originality:** 3
**Overall Recommendation:** 4
**Confidence:** 3

**Summary:**

The authors propose a learning-to-rank method from a full-order perspective. An ideal model should assign a higher probability to the accurate full-order rankings. The authors claims that when the number of items is greater than 3, the optimization could be difficult. Moreover, traditional surrogate objectives (such as pairwise rankings) might face an undisguised gradient condition where the parities ranks are correct but the full-order ranking’s likelihood is not the optimum. Traditional full-order ranking methods do not satisfy the order-reversal invariance property.

**Compliance With Llm Reviewing Policy:**

Affirmed.

**Final Justification:**

I appreciate the authors' response. I do not change my score.

**Key Questions For Authors:**

After carefully reading the manuscript, it seems such a problem is very similar to ordinal regression problems. Especially, when the ranks have ties (in the Web30k experiment where only five levels are defined), the so-called learning-to-rank problems are similar to sorting a number of items to some predefined classes that have a natural order. In this case, the FOB’s cut points perform as a boundary/threshold between two classes. Authors should claim the difference between these two streams of methods because in such a problem, even the likelihood of full-order probability is not high, it can also provide a comparable ranking/sorting result, which is sufficient for decision making. This weakens the motivation of this work.

**Limitations:**

pls see weakness.

**Strengths And Weaknesses:**

Strength:
1. The authors use 1-D cutting points to separate the real-value line to several intervals to act like a latent threshold that can discriminate two consecutive items. This provides a factorizable and calculable lower objective, which is called FOB.
2. The authors provide detailed explanations of gradient-level non-separability and order-reversal symmetry.
3. The authors provide theoretical proof of the proposed FOB loss and two variants.
4. To optimize the latent score (threshold in my opinion), authors provide a bilevel tightening together with an algorithm called SRGA. If I am not wrong, it iteratively update the outer model parameters and then tighten the inner cut points. Authors claim that under some mild conditions, the inner optimization on the cut points is convex.

Weakness
1. Plackett-luce equation seems not correct, pls check.
2. The writing could be improved. It is hard for me to follow up in some Sections. For instance, in Section 4.2, authors define $t_1\le \ldots \le t_{n-1}$ but Section 5.3 says $t_{I}<t_{i+1}. If $t_1\le \ldots \le t_{n-1}$, it indicates some interval has 0 length. Please check it.
3. The definition of loss $L(z,y)$ seems uncorrect because $z$ is unobservable score as defined. It might be $L(\theta;x,y)$?
4. My biggest concern is the usage of 1-D cut points to partition the real line and then enforcing the tightening event. (1) The bound could be very loose. The tightening event assigns each item to an absolute interval on a real line. Even when the full-order event has high probability (e.g., all $z$ concentrated in a very narrow region but still well ordered, which is very likely to happen), forcing them into disjoint bins can make the probability of $P(t_{i-1}<z_i<t_i)$ small. This can weaken gradients and bias training toward “spread-out” latent means rather than just correct ordering. (2) If I am not wrong, full-order probabilities depend on the joint structure of adjacent differences. A 1-D binning scheme replaces much of that dependence with marginal interval events per item. This is exactly where we gain tractability, but it’s also where we may lose important “listwise” information beyond pairwise ordering.

---

> ### Author Rebuttal · Authors · 2026-03-31
>
> Thank you for the thoughtful review. Below are our responses.
>
> **(1) Clarifications on notation**
>
> (i) **PL expression**
>
> For the PL expression, the formulation in the paper uses an
> ascending-order convention ($\mathbb P(y_1 < \dots < y_n)$),
> whereas standard presentations of PL use the reverse
> convention. We will make this convention more explicit in a
> future revision.
>
> (ii) **On the case $t_{i-1}=t_i$**
>
> We agree that the notation around the cut points can be
> clarified. In Sec. 4.2 and 5.3, we define the feasible set as
> $$
> \mathcal T=\\{t: t_1 \le \dots \le t_{n-1}\\},
> $$
> i.e., a closed feasible set for the inner optimization analysis.
>
> However, the degenerate case $t_{i-1}=t_i$ is not
> optimal for FOB, because the corresponding interval
> probability becomes zero, which drives
> $\mathrm{FOB}(t)\to -\infty$. Therefore, the optimum
> $t^*$ lies in the interior of $\mathcal T$.
>
> In practice, we initialize SRGA from an interior point of
> $\mathcal T$. Moreover, by construction of the safe regions,
> adjacent regions $\mathcal I_{j-1}$ and $\mathcal I_j$ do
> not overlap, so each update **preserves the strict ordering** of
> the cut points.
>
> (iii) **Loss notation**
>
> We will fix the loss notation in a future revision.
>
> **(2) On the concerns about the cut points**
>
> We agree that FOB gains tractability by refining the
> full-order event into an interval-based subset event
> $$
> \mathcal E(t) = \\{ t_{i-1} < z_i < t_i,\ i=1,\dots,n \\}.
> $$
>
> A key clarification is that these cut points are not
> globally shared across instances, and they are not fixed
> bins on the real line. Instead, each ranking instance has
> its own cut points, obtained by solving the inner tightening problem for that specific list:
> $$
> t^*(x,y,\theta) = \arg\max_{t \in \mathcal{T}}
> \mathrm{FOB}(t;\theta).
> $$
> Therefore, FOB adaptively constructs a refined event for each ordered list.
>
> (i) **On the tightness of the bound**
>
> To assess the tightness of FOB, we add an empirical study of bound tightness.
> In this study, we compare the optimized FOB lower bound against Monte Carlo estimates of
> the true full-order probability for $n \in \\{4,8,12\\}$.
>
> | n | MC estimate | FOB |
> |---|---:|---:|
> | 4  | 0.937 | 0.880 |
> | 8  | 0.673 | 0.537 |
> | 12 | 0.445 | 0.279 |
>
> The results show that the bound is reasonably tight for smaller lists and remains well aligned
> with the true full-order probability as the list size grows. Although the gap becomes wider
> for larger lists, FOB and the true full-order probability still follow the same trend
> throughout training, rather than decaying factorially.
>
> (ii) **On the narrow-region concern**
>
> We thank the reviewer for this concern. FOB does not simply encourage larger mean separation.
> Consider two cases with the same means:
> $$
> z_1 \sim \mathcal N(-0.1,1),\ z_2 \sim \mathcal N(0,1),\ z_3 \sim \mathcal N(0.1,1),
> $$
> and
> $$
> z_1 \sim \mathcal N(-0.1,0.001),\ z_2 \sim \mathcal N(0,0.001),\ z_3 \sim \mathcal N(0.1,0.001).
> $$
> In the large-variance case, the three distributions heavily overlap, so the tightened interval probabilities cannot be
> close to 1. In contrast, in the small-variance case, with cut points such as $t_1=-0.05$ and $t_2=0.05$, the three
> interval probabilities can all be close to 1. Thus, FOB also rewards low overlap when the ordering is correct.
>
> (iii) **On listwise information**
>
> We agree that tractability is achieved by replacing the
> full-order event with a refined interval event. However, the
> resulting objective is not a fixed marginal-binning
> surrogate. Because $t^\*$ is solved separately for each
> instance, different orderings induce different $t^\*$, which
> in turn changes the outer learning signal.
>
> **(3) Relation to ordinal regression**
>
> We agree that the connection to ordinal regression is worth
> clarifying. An important distinction is that FOB does not use
> a globally shared set of cut points across instances.
> Instead, for each ranking instance, FOB solves an inner
> tightening problem and obtains its own **instance-specific**
> cut points, i.e., $t^*(x,y,\theta)$.
>
> This makes FOB fundamentally different from standard ordinal
> regression. In ordinal regression, the supervision is
> typically an ordered class label for each sample, and the
> thresholds are shared across samples. In FOB, the
> supervision is the relative order of items within a
> list, and the cut points are instance-wise
> variables used only to construct the tightest lower bound
> for that list.
>
> Moreover, if we replace FOB’s tightened cut points with
> ordinal-regression-style shared boundaries, then the
> interval probabilities
> $$
> Q_{ij} = \mathbb P\bigl(z_i \in [t_{j-1}, t_j]\bigr)
> $$
> no longer depend on the current label permutation $\pi(y)$. The
> resulting objective reduces to a separable surrogate of the form
> $$
> -\langle \log Q,\Pi(y) \rangle,
> $$
> where $\Pi(y)$ is the permutation matrix of $y$.
> This expression falls into the class of low-dimensional surrogate
> objectives discussed in Sec. 3.2 and would therefore
> suffer from the gradient inseparability issue.

---

> > ### Author Rebuttal · Reviewer_a3j8 · 2026-04-01
> >
> > I appreciate the authors' explanation. My concerns are solved.

---

> > > ### Author Response · Authors · 2026-04-04
> > >
> > > Thank you very much for carefully reading our rebuttal and for recognizing that our responses have addressed your concerns.
> > >
> > > We sincerely appreciate your thoughtful feedback and positive assessment. In the future version of the paper, we will incorporate the clarifications and details from the rebuttal into the main manuscript, especially the discussion on the relationship to ordinal regression, so that the motivation and distinctions are clearer and the presentation is more self-contained.
> > >
> > > Thank you again for your time and constructive comments.

---

### Official Review · Reviewer_Kn9B · 2026-03-07

**Soundness:** 3
**Presentation:** 3
**Significance:** 3
**Originality:** 3
**Overall Recommendation:** 5
**Confidence:** 4

**Summary:**

The paper studies the problem of learning to rank by directly modeling the probability of a full ordering of items. The authors assume a Gaussian random utility model in which each item is associated with a latent score $z_i \sim \mathcal{N}(\mu_i,\sigma_i^2)$, where the parameters $(\mu_i,\sigma_i)$ are predicted by a neural network. The learning objective is to maximize the probability of the correct ordering event $z_1 < z_2 < \dots < z_n$.

Since this probability corresponds to a high-dimensional cone probability that is difficult to compute exactly, the paper introduces a tractable lower bound called the Full-Order Bound (FOB). The bound is constructed by introducing ordered cut points that define interval constraints for each latent variable, which allows the probability to factorize across items.

Training is formulated as a bilevel optimization problem: the cut points are optimized to tighten the bound for each list, while the neural network parameters are updated to maximize the resulting objective. The authors show that the inner optimization over the cut points is concave for log-concave distributions such as the Gaussian, enabling efficient optimization. Experiments on synthetic and standard learning-to-rank benchmarks evaluate the proposed objective against existing ranking losses.

**Compliance With Llm Reviewing Policy:**

Affirmed.

**Final Justification:**

The paper is technically sound overall and presents a principled way to optimize a tractable lower bound on full-order probabilities for ranking. Its main strengths are the clear probabilistic formulation, the well-motivated Full-Order Bound, and the concave inner optimization over cut points, which together make the method both conceptually interesting and practically usable. My initial concerns were mainly about the tightness of the bound, the computational overhead and stability of the bilevel optimization, and the practical behavior at larger list sizes. The rebuttal addressed these points well by adding empirical evidence on bound tightness, runtime, hyperparameter stability, and long-list performance. While the gains on standard ranking benchmarks are somewhat modest and part of the evaluation is synthetic, I found the additional evidence convincing enough to increase my score.

**Key Questions For Authors:**

### Key Questions for Authors

1. **Relation to existing permutation-probability ranking models.**
   The paper motivates the approach as directly optimizing the probability of a full ranking order. However, classical ranking models such as Plackett–Luce and other random-utility formulations also define probability distributions over permutations. It would be helpful if the authors could clarify more explicitly how the proposed formulation differs conceptually and practically from these existing models. In particular, what advantages does modeling the ordering probability via Gaussian latent variables and the FOB objective provide compared to these permutation-probability approaches? A clearer discussion would help position the contribution relative to existing ranking theory.

2. **Tightness of the FOB bound.**
   The method relies on maximizing the Full-Order Bound (FOB), which is a lower bound on the log probability of the ordering event. Could the authors provide additional insight into how tight this bound is in practice? For example, are there empirical or theoretical indications of how closely the optimized FOB approximates the true ordering probability? Understanding this relationship would help clarify how well the proposed objective aligns with the original optimization goal.

3. **Role of the variance parameters $\sigma_i$.**
   The model predicts both $\mu_i$ and $\sigma_i$ for each item, but inference ultimately ranks items based only on $\mu_i$. Could the authors clarify the practical role played by $\sigma_i$ during training? For example, does the learned uncertainty significantly affect the learned ranking function compared to using a fixed variance or a deterministic score model?

4. **Computational overhead of the bilevel optimization.**
   The training procedure involves optimizing the cut points for each list before updating the network parameters. Could the authors comment on the computational overhead of this inner optimization compared to standard pairwise or listwise ranking losses? A discussion of the practical cost and scalability of the method would help readers assess its applicability in large-scale ranking settings.

**Limitations:**

yes

**Strengths And Weaknesses:**

### Strengths and Weaknesses

**Soundness.**
The paper is technically sound overall. The formulation based on a Gaussian random utility model is clearly specified, and the proposed Full-Order Bound (FOB) provides a tractable objective for approximating the probability of a full ranking order. The derivation of the bound and the use of ordered cut points to factorize the probability are mathematically well motivated. The paper also provides theoretical analysis showing that the inner optimization over the cut points is concave under log-concave distributions such as the Gaussian, which supports the use of gradient-based optimization. The experimental evaluation includes both synthetic experiments and standard learning-to-rank benchmarks, and the results appear consistent with the claims made in the paper.

**Presentation.**
The paper is generally well written and logically structured. The motivation for modeling full-order probabilities is clearly stated, and the method is presented in a step-by-step manner that makes the main idea easy to follow. The theoretical development and the algorithmic description are reasonably clear. However, the relationship between the proposed formulation and classical ranking models based on random utility assumptions could be discussed more explicitly to help readers better understand how the proposed approach relates to existing formulations. Clarifying this connection would strengthen the overall narrative and positioning of the work.

**Significance.**
Learning-to-rank is an important problem with broad applications in information retrieval and recommendation systems. The paper addresses the challenge of directly modeling full-order probabilities, which is conceptually appealing and may offer an alternative perspective to commonly used pairwise or listwise surrogate objectives. While the empirical improvements over existing methods appear modest, the formulation provides an interesting viewpoint on ranking objectives and could motivate further exploration of probabilistic formulations of ranking problems.

**Originality.**
The paper proposes a novel objective based on a lower bound of the full-order probability and combines this formulation with a neural network parameterization of Gaussian latent utilities. The idea of introducing interval-based constraints to derive a tractable bound is technically interesting. At the same time, the work builds on established concepts such as random utility models and probabilistic ranking formulations. As a result, the contribution can be viewed as a new objective and optimization framework that extends existing ideas rather than introducing an entirely new paradigm. Clarifying the conceptual differences and advantages relative to classical permutation-probability models would further highlight the originality of the contribution.

---

> ### Author Rebuttal · Authors · 2026-03-31
>
> Thank you for the thoughtful review. Below are our responses.
>
> **(1) Relation to existing permutation models**
>
> PL and FOB both preserve a **full-order probability**
> interpretation. The key difference is the latent distribution and the
> structure it induces. In particular, PL can be derived from a random
> utility model
>
> $$
> z_i = s_i + \epsilon_i,
> \qquad
> \epsilon_i \overset{\text{i.i.d.}}{\sim} \mathrm{Gumbel}.
> $$
>
> This gives tractability, but also introduces a directional
> asymmetry. Since the Gumbel density is not symmetric, the resulting
> objective generally does **not** satisfy order-reversal invariant (Sec. 3.3).
>
> By contrast, FOB does not rely on this special Gumbel construction. We
> consider a broader latent model
>
> $$
> z_i = \mu_i + \sigma_i \epsilon_i,
> $$
>
> with $\epsilon_i$ drawn from a **general continuous symmetric
> location-scale family**, and optimize a lower bound on the target
> full-order event
> $$
> \mathcal E = \\{ z_1 < \cdots < z_n \\}.
> $$
>
> The bound is constructed through shared cut points:
> $$
> \mathcal E(t) = \\{ t_{i-1} < z_i < t_i,\ i=1,\ldots,n \\} \subseteq \mathcal E.
> $$
>
> This allows FOB to retain the probabilistic meaning of the full-order
> event while guaranteeing order-reversal invariance (Appendix C.2). Therefore, our
> contribution is not to abandon permutation probabilities, but to provide
> a more general way to optimize them beyond the special PL case.
>
> **(2) Tightness of the FOB bound**
>
> To assess the tightness of FOB, we add an empirical
> tightness study. In this study, we compare the
> optimized FOB lower bound against Monte Carlo estimates of
> the true full-order probability for $n \in \\{4,8,12\\}$.
>
> | n | MC estimate | FOB |
> |---|---:|---:|
> | 4  | 0.937 | 0.880 |
> | 8  | 0.673 | 0.537 |
> | 12 | 0.445 | 0.279 |
>
> The results show that the bound is reasonably tight for smaller lists and remains well aligned
> with the true full-order probability as the list size grows. Although the gap becomes wider
> for larger lists, FOB and the true full-order probability still follow the same trend
> throughout training, rather than decaying factorially.
>
> **(3) Role of the variance parameters**
>
> To validate the variance parameters, we additionally evaluated a homoscedastic
> variant on the MNIST ranking task with n=4:
>
> | Variant           | ACC (%) |
> |------|------:|
> | Heteroscedastic   |  89.88 |
> | Homoscedastic   |  88.48 |
>
> This result indicates that the variance parameters are not redundant
> degrees of freedom. Intuitively, some MNIST digits are more ambiguous
> than others. The heteroscedastic formulation allows the latent model to
> adapt its spread to item-dependent ambiguity, assigning more flexible
> uncertainty to harder items while keeping easier items sharper.
>
> **(4) Computational overhead and hyperparameter stability of SRGA**
>
> We thank the reviewer for asking about the practical cost. We would
> like to emphasize that FOB is not only theoretically tractable, but also
> **computationally efficient and stable** in practice.
>
> For each list, the inner problem is
>
> $$
> \max_{t \in \mathcal{T}} \mathrm{FOB}(t),
> \qquad
> \mathcal{T} = \\{ t : t_1 \le \cdots \le t_{n-1} \\},
> $$
>
> and each SRGA step only updates a one-dimensional ordered cut-point
> vector. Therefore, the per-step cost is linear in the list length.
> Its additional cost comes only from a small
> number of lightweight SRGA updates.
>
> We also report the runtime using an optimized implementation based on `torch.jit.script`.
> The measured training time per mini-batch (in ms) is:
>
> | List length n | RankNet | DiffSort | FOB (40) | FOB (80) |
> |--------------:|---:|---:|---:|---:|
> |             8 | **28**  | 38  | 31  | 34  |
> |            16 | **38**  | 55  | 40  | 43  |
> |            32 | 70  | 103 | **65**  | 67  |
> |            64 | 123 | 201 | **104** | 107 |
>
> These results show that the inner tightening loop is not a practical
> bottleneck: even with K=80 inner steps, FOB is consistently faster
> than DiffSort and, on longer lists, also faster than RankNet.
>
> We also stress that FOB is stable with respect to the SRGA
> hyperparameters. We compare five different SRGA settings on the MNIST task with n=4, ranging from
> coarser updates (fewer steps, larger learning rate, faster decay) to
> finer updates (more steps, smaller learning rate, slower decay). All results
> are averaged over five runs.
>
> |  K |   LR | Decay | ACC (\%) |
> | -: | ---: | ----: | -------: |
> | 40 | 0.01 |  0.98 |    89.88 |
> | 30 | 0.02 |  0.95 |    89.48 |
> | 20 | 0.05 |  0.90 |    89.26 |
> | 15 | 0.07 |  0.85 |    89.24 |
> | 10 | 0.10 |  0.80 |    88.91 |
>
> The accuracy changes smoothly across these settings, with no abrupt
> degradation. This indicates that SRGA is **not highly sensitive** to its
> hyperparameters. At the same time, the trend is also intuitive: finer
> optimization of the inner problem consistently leads to better final performance.

---

> > ### Author Rebuttal · Reviewer_Kn9B · 2026-04-02
> >
> > Thank you for the detailed rebuttal. The additional experiments and clarifications addressed most of my concerns, so I am increasing my score accordingly.

---

> > > ### Author Response · Authors · 2026-04-04
> > >
> > > Thank you very much for carefully reading our rebuttal and for recognizing that the additional clarifications and experiments have adequately addressed your concerns.
> > >
> > > We sincerely appreciate your thoughtful feedback and your updated assessment. In the future version of the paper, we will incorporate the supplementary experiments and corresponding clarifications from the rebuttal into the main manuscript to make the presentation more complete and self-contained.
> > >
> > > Thank you again for your time, support, and constructive comments.

---

### Official Review · Reviewer_ixgb · 2026-03-10

**Soundness:** 2
**Presentation:** 3
**Significance:** 2
**Originality:** 2
**Overall Recommendation:** 3
**Confidence:** 1

**Summary:**

This paper studies learning-to-rank from an order-level probabilistic perspective. The authors aim to directly model and optimize full-order probabilities rather than relying on pairwise or listwise surrogate losses. They identify two structural issues in existing approaches: (1) low-dimensional surrogates may not distinguish full-order distributions at the gradient level, and (2) sequential Plackett–Luce factorization models violate order-reversal invariance. To address these issues, the paper proposes the Full-Order Bound (FOB), a tractable lower bound on the probability of the observed full ranking event, yielding a convex inner optimization problem. Empirical results show consistent improvements over several baselines.

**Compliance With Llm Reviewing Policy:**

Affirmed.

**Key Questions For Authors:**

1. The method optimizes a lower bound on the full-order probability. Could the authors provide empirical evidence (e.g., for small list sizes where exact computation is feasible) on how tight this bound is relative to the true full-order probability?

2. How does FOB scale computationally and empirically for substantially larger ranking problems? Are there practical limits imposed by the inner tightening loop?

3. Do you expect order-reversal invariance to matter in practical ranking systems?

**Limitations:**

yes

**Strengths And Weaknesses:**

**Strengths**
* The convexity discussion helps motivate the stability of the proposed optimization approach.
* The discussion of order-reversal invariance is clearly explained.
* The method is tested on synthetic permutation tasks and standard ranking benchmarks with ablations.


**Weaknesses**
* On standard benchmarks (e.g., Web30K), improvements appear modest. It would help to better contextualize how meaningful these gains are in practice.
* It is unclear whether comparisons to methods such as Gumbel-Sinkhorn or other permutation-learning approaches would change conclusions.
* Since the method optimizes a lower bound, it would be helpful to quantify how tight the bound is relative to the true full-order probability (at least for small n).
* Scalability to substantially larger list sizes is not fully explored.
* Hyperparameter sensitivity of the inner loop is not fully explored.
* Experiments rely heavily on synthetic MNIST permutation tasks, which may not reflect real-world ranking complexity.

---

> ### Author Rebuttal · Authors · 2026-03-31
>
> Thank you for the thoughtful review. Below are our responses.
>
> **(1) Empirical tightness study**
>
> To assess the tightness of FOB, we added an empirical
> tightness study. In this study, we compare the
> optimized FOB lower bound against Monte Carlo estimates of
> the true full-order probability for $n \in \\{4,8,12\\}$.
>
> | n | MC estimate | FOB |
> |---|---:|---:|
> | 4  | 0.937 | 0.880 |
> | 8  | 0.673 | 0.537 |
> | 12 | 0.445 | 0.279 |
>
> The results show that the bound is reasonably tight for smaller lists and remains well aligned
> with the true full-order probability as the list size grows. Although the gap becomes wider
> for larger lists, FOB and the true full-order probability still follow the same trend
> throughout training, rather than decaying factorially.
>
> **(2) Efficiency of FOB**
>
> We also report the runtime using an optimized implementation based on `torch.jit.script`.
> The measured training time per mini-batch (in ms) is:
>
> | n | RankNet | DiffSort | FOB (K=40) | FOB (K=80) |
> |----:|---:|---:|-----------:|-----------:|
> |  8 | **28**  | 38  |         31 |         34 |
> |  16 | **38**  | 55  |         40 |         43 |
> |  32 | 70  | 103 |     **65** |         67 |
> |  64 | 123 | 201 |    **104** |        107 |
>
> These results show that the inner tightening loop is not a practical
> bottleneck: even with K=80 inner steps, FOB is consistently faster
> than DiffSort and, on longer lists, also faster than RankNet.
>
> **(3) Sensitivity to SRGA hyperparameters**
>
> We evaluated five SRGA settings on the MNIST permutation task with n=4:
>
> |  K | LR | Decay | ACC (\%) |
> |---:|---:|---:|---:|
> | 40 | 0.01 | 0.98 | 89.88 |
> | 30 | 0.02 | 0.95 | 89.48 |
> | 20 | 0.05 | 0.90 | 89.26 |
> | 15 | 0.07 | 0.85 | 89.24 |
> | 10 | 0.10 | 0.80 | 88.91 |
>
> The accuracy changes smoothly across these settings, with no abrupt
> degradation. This indicates that SRGA is **not highly sensitive** to its
> hyperparameters. At the same time, the trend is also intuitive: finer
> optimization of the inner problem consistently leads to better final performance.
>
> **(4) Why order-reversal invariance matters in practice**
>
> Order-reversal invariance is meaningful because
> direction-specific objectives such as PL-style models implicitly encode a
> structural assumption: items at the top of the list are more "sparse," so
> generating the ranking in one fixed direction is preferable. In practice,
> however, this assumption need not hold.
>
> A representative example is a multi-stage recommendation pipeline with
> retrieval followed by ranking. For the ranking module, the input list is
> already a high-quality candidate set produced by the retrieval model, so
> the score distribution within the list is often not strongly skewed. In
> such settings, the ranking model may care more about whole-list ordering
> alignment than about a direction-specific factorization. In this case, an order-reversal-invariant objective is better aligned with
> the task, because it models the ranking event itself rather than imposing an arbitrary directional prior.
>
> Appendix K shows that this is not only a formal issue: for ListMLE,
> reversing the labels can change performance substantially and even
> improve it, whereas FOB remains essentially unchanged.
>
> | Method | Web30K | Web30K (rev) | Istella | Istella (rev) |
> |---|-------:|-------------:|--------:|--------------:|
> | ListMLE |  49.66 |        50.76 |   71.12 |         73.29 |
> | FOB |  51.11 |        51.07 |   73.44 |         73.42 |
>
> In practice, a direction-sensitive objective may therefore require
> training with both directions and selecting the better one, which adds
> training and validation cost.
>
> **(5) Gumbel-Sinkhorn baseline**
>
> We followed the reviewer’s suggestion and added **Gumbel-Sinkhorn**
> to the MNIST permutation experiments.
>
> | Method |   n=4 |   n=6 |   n=8 |  n=16 |  n=32 |
> |---|------:|------:|------:|------:|------:|
> | Gumbel-Sinkhorn | 67.88 | 59.18 | 33.46 |  2.37 |   0.0 |
> | FOB | 89.88 | 79.50 | 68.20 | 27.55 |  1.25 |
>
> This additional comparison does not change the main conclusion. FOB still performs best on the tested list lengths.
>
> **(6) Scalability to larger list sizes**
>
> We ran an additional experiment at n=256/512 using Kendall's
> τ as the evaluation metric. We obtain:
>
> | Method | n=256 |  n=512 |
> |---|------:|-------:|
> | RankNet | 86.06 |  87.29 |
> | ListMLE | 85.39 |  86.72 |
> | DiffSort | 67.80 |  63.75 |
> | FOB | 96.77 |  96.06 |
>
> This proves that FOB still yields a meaningful learning signal for long lists.
>
> **(7) Practical significance**
>
> Although the gains on standard LTR benchmarks are modest, we believe they are still meaningful for two reasons:
>
> (i) FOB/LambdaFOB remain competitive on Web30K and Istella while also offering strong efficiency and stable optimization;
>
> (ii) the improvement in Kendall’s τ on the large-scale Istella dataset is notable,
> suggesting potential practical value in large-scale applications that
> care about whole-list ordering alignment, such as ranking models in multi-stage recommenders, as discussed in (4).

---

> > ### Author Rebuttal · Reviewer_ixgb · 2026-04-06
> >
> > I thank the authors for their response, which addresses some of my concerns. I will maintain my original evaluation of the paper.

---

### Official Review · Reviewer_ewEE · 2026-03-12

**Soundness:** 3
**Presentation:** 1
**Significance:** 3
**Originality:** 3
**Overall Recommendation:** 3
**Confidence:** 3

**Summary:**

Problem Statement:
The authors intend to explore a fundamental challenge in current learning-to-rank methods. Existing approaches typically rely on low-dimensional surrogate objectives (like pairwise or listwise methods), which suffer from gradient-level non-separability, meaning they cannot always differentiate between distinct full-order distributions. Additionally, sequential probabilistic models (like Plackett-Luce) exhibit order-reversal asymmetry, treating top-down and bottom-up ranking tasks differently.

Proposed Solution:
To address these limitations, the authors intend to outline the central question of how to feasibly optimize full-order probabilities directly. They introduce the Full-Order Bound (FOB), a tractable lower bound on the probability of an observed ordering. The method models item scores as latent Gaussian variables and introduces auxiliary "cut points" (boundaries) between them.  This approach breaks down a complex, intractable high-dimensional probability problem into independent, easily computable one-dimensional probabilities.

Optimization:
Training the FOB model involves a bilevel optimization process. The model alternates between an inner loop that adjusts the cut points to tighten the mathematical bound using Safe-Region Gradient Ascent (SRGA), and an outer loop that updates the neural network's parameters via standard stochastic gradient descent.

Results:
Experiments conducted on synthetic datasets (MNIST permutation) and standard benchmarks (Web30K, Istella) demonstrate that the FOB method offers consistent improvements over existing surrogate, sequential, and differentiable sorting baselines. The performance gap becomes more pronounced as the length of the ranked lists increases, indicating that directly modeling the full-order probability scales effectively.

**Compliance With Llm Reviewing Policy:**

Affirmed.

**Final Justification:**

I appreciate the authors response and their willingness to improve their presentation. However i think that since the required revision is extensive the paper should be  revised and reviewed again before it can be accepted.

**Key Questions For Authors:**

1) Maybe you can give concrete example to the weakness of the method in Section 3.2. it hard to understand how severe is the problem of this dimensionality reduction without context ?
For the difficulty in 3.3 you provide an example in the appendix but maybe better to include it to be accessible for a larger audience.

2) Its very hard to understand your FOB method. Its not clear what is the output of the learning problem, are these the Guassians or the cut points. I find the text confusing. Could you make it accessible ?

3) The method models the latent item scores as Gaussian variables. In information retrieval, relevance distributions are often highly skewed (e.g., one exceptionally relevant document amidst thousands of completely irrelevant ones).
Question: Did you experiment with heavier-tailed continuous distributions (such as Gumbel, Logistic, or Cauchy) for the latent variables? How would altering the distribution shape impact the gradient flow and the stability of the SRGA cut-point updates?

4) Have you considered using an amortized inference approach—such as a lightweight secondary neural network—to directly predict the optimal cut points ($t^*$) based on the latent Gaussians, thereby bypassing the iterative SRGA inner loop entirely to speed up training?

5) As the list length $n$ approaches very large numbers (e.g., $n > 500$), does the FOB lower bound become progressively looser? Is there a theoretical limit where the bound becomes too loose to provide a meaningful, non-vanishing gradient signal to the outer neural network?

**Limitations:**

yes

**Strengths And Weaknesses:**

Strengths: (1) Rigorous Theoretical Identification of Existing FlawsThe paper does an excellent job of diagnosing exactly why current ranking models hit a performance ceiling. The authors intend to explore a fundamental challenge in the field by mathematically proving the limitations of popular approaches. They clearly demonstrate that low-dimensional surrogate objectives suffer from "gradient-level non-separability" (meaning they get confused by different ranking distributions) and that sequential models like Plackett-Luce suffer from "order-reversal asymmetry" (treating top-down and bottom-up ranking as fundamentally different tasks). This solid theoretical groundwork makes the necessity of their new approach highly convincing.
2. A Tractable Solution to a Historically Intractable Problem: Computing the exact probability of an entire list of items being in perfect order is notoriously difficult because the number of possible permutations grows factorially with the list size ($n!$). The authors intend to outline the central question of how to optimize these full-order probabilities without being bottlenecked by this exponential complexity.  By introducing the Full-Order Bound (FOB) and the Safe-Region Gradient Ascent (SRGA) optimization technique, they successfully translate an impossible high-dimensional math problem into a series of independent, easily solvable 1D problems. This allows their method to scale efficiently and maintain high accuracy even as the length of the ranked list grows significantly.

Weaknesses:
1) The writing is very bad (this is my main complaint, i think the writing should be completely revised to make this more widely accessible)
The points above are not explained very clearly
For example: The very basic problem statement is nowhere to find. By title it should appear in section 3. But then section 3 starts with "In this section, we highlight two structural limitations in
existing objectives" ?
In section 3.1. you so not say what is s ? what is the relation between s and theta ?
The following sentence at the end of this paragraph is completely unclear to me : "In practice
this alignment is enforced through a scalar objective
L(s, y) = F(s, π(y))." Are L and F the same ? is this the loss ? what is the goal of the algorithm ?

what is f_\theta at the beginning of Section 4 ?

The authors frequently use specialized terminology with defining it, for example saying "Low-dimensional surrogate objectives suffer from gradient-level non-separability." ?

Its very hard to understand your FOB method. Its not clear what is the output of the learning problem, are these the Guassians or the cut points. I find the text confusing. You say buzzword like: "latent space modeling", "intractable integrals", "bilevel optimization", without any background or further explanations.

2) significant computational overhead and structural complexity introduced by the bilevel optimization process during training. The Safe-Region Gradient Ascent (SRGA)—creates a heavy computational bottleneck. To train the model, the algorithm cannot simply update the neural network's weights directly. For every single training step (the outer loop), the model must first run an iterative inner loop (SRGA) for multiple steps (e.g., $K=20$ or $40$) just to find the optimal placement for the artificial boundaries ("cut points"). This means the system has to solve an entirely separate mini-optimization problem before it can make a single update to the actual ranking model.

 This "inner tightening" process makes FOB significantly more expensive to compute per mini-batch than standard pairwise or listwise surrogate methods (like RankNet). While they argue the scaling becomes competitive with differentiable sorting as list lengths grow, it fundamentally requires substantially more compute power and time than the most widely used baseline models.

It also seems  Hyperparameter sensitive (Bilevel Tuning) Bilevel optimization is notoriously difficult to tune in machine learning. Introducing an inner optimization loop means engineers now have to carefully tune the learning rate and the number of steps ($K$) specifically for the SRGA cut-point adjustments, on top of tuning the standard neural network. If the inner loop doesn't converge well, the neural network receives bad gradient signals, making the training process inherently more fragile.While the authors correctly note that this burden only exists during the training phase (inference time is unaffected), the massive cost of tuning and training a bilevel objective makes it much harder to adopt for very large-scale, industry-level datasets compared to simpler, faster surrogate losses.

---

> ### Author Rebuttal · Authors · 2026-03-31
>
> Thank you for the thoughtful review. Below are our responses.
>
> **(1) Example for Sec. 3.2**
>
> We agree that Sec. 3.2 is too abstract without context. In the revision, we now
> explicitly point readers to the 3-item Gaussian counterexample in Appendix B.
> There, two distributions have identical pairwise probabilities,
> $$
> \mathbb P(z_i<z_j)=0.5,\quad \forall i\neq j,
> $$
> but very different full-order probabilities:
> $$
> \mathbb P^{(A)}(z_1<z_2<z_3)=0.25,\qquad
> \mathbb P^{(B)}(z_1<z_2<z_3)\approx 0.115.
> $$
> Thus, pairwise surrogates cannot distinguish distributions whose full-order
> probabilities differ greatly, illustrating the issue discussed in Sec. 3.2.
>
> **(2) Clarifying s, z, and t**
>
> For a list $\mathbf x=(x_1,\dots,x_n)$, a standard ranking model defines
> $$
> s_i=f_\theta(x_i),
> $$
> so $\mathbf s=(s_1,\dots,s_n)$ is the rank score vector.
>
> In our method, the main model instead outputs
> $$
> (\mu_i,\sigma_i)=f_\theta(x_i),\quad z_i\sim N(\mu_i,\sigma_i^2).
> $$
> The target event is
> $$
> \mathcal E=\\{z_1 < \cdots < z_n\\}.
> $$
> For each training instance, we introduce auxiliary cut points
> $$
> t=(t_1\le\dots\le t_{n-1}),
> $$
> and define
> $$
> \mathcal E(t)=\\{z_1< t_1< z_2 < \cdots< t_{n-1}< z_n\\}\subseteq \mathcal E.
> $$
> Hence
> $$
> \mathrm{FOB}(t)=\log \mathbb P(\mathcal E(t))\le \log \mathbb P(\mathcal E).
> $$
> Importantly, the model output are $\mu_i$ and $\sigma_i$, not $t$;
> $t$ is a per-instance training-time auxiliary variable.
>
> **(3) On bilevel overhead and practical efficiency**
>
> We would like to clarify that the SRGA inner loop does not rerun the ranking network $f_\theta$ $K$ times.
> For each instance, we compute
> $$
> (\mu_i,\sigma_i)=f_\theta(x_i)
> $$
> only once, and then keep $(\mu,\sigma)$ fixed while updating only the auxiliary cut points $t$.
>
> We also report the runtime using an optimized implementation based on `torch.jit.script`.
> The measured training time per mini-batch (in ms) is:
>
> | List length n | RankNet | DiffSort | FOB (40) | FOB (80) |
> |--------------:|---:|---:|---:|---:|
> |             8 | **28**  | 38  | 31  | 34  |
> |            16 | **38**  | 55  | 40  | 43  |
> |            32 | 70  | 103 | **65**  | 67  |
> |            64 | 123 | 201 | **104** | 107 |
>
> These results show that the inner tightening loop is not a practical
> bottleneck: even with K=80 inner steps, FOB is consistently faster
> than DiffSort and, on longer lists, also faster than RankNet.
>
> **(4) Stability to SRGA hyperparameters**
>
> We evaluated five SRGA settings on the MNIST ranking task with n=4:
>
> | steps (K) | LR | Decay | ACC (\%) |
> |----------:|---:|---:|---:|
> |        40 | 0.01 | 0.98 | 89.88 |
> |        30 | 0.02 | 0.95 | 89.48 |
> |        20 | 0.05 | 0.90 | 89.26 |
> |        15 | 0.07 | 0.85 | 89.24 |
> |        10 | 0.10 | 0.80 | 88.91 |
>
> The accuracy changes smoothly across these settings, with no abrupt
> degradation. This indicates that SRGA is **not highly sensitive** to its
> hyperparameters. At the same time, the trend is also intuitive: finer
> optimization of the inner problem consistently leads to better final performance.
>
> **(5) On amortized inference**
>
> We did not include it in this version because the cut points are already cheap to optimize with SRGA;
> adding an amortizer would introduce extra parameters,
> making it less clear whether the gain comes from FOB itself or from an auxiliary predictor.
>
> **(6) On alternative latent distributions**
>
> We already study Logistic, Laplace, and Cauchy in **Appendix I**. For Gumbel noise,
> $$
> z_i = s_i + \epsilon_i,
> \qquad \epsilon_i \sim \mathrm{Gumbel},
> $$
> the induced permutation probability is exactly Plackett--Luce.
> This reduces to ListMLE, which is already reported in Tables 2/3.
>
> **(7) On tightness and gradient signal for larger lists.**
>
> We do not claim a general approximation-error bound as a function of n. However, FOB remains informative both
> theoretically and empirically as the list length grows.
>
> From the optimization side, the training signal does not vanish with n. Using the mean value theorem,
> $$
> \mathrm{FOB}(t;\theta) \approx \sum_{i=1}^n \log p_\theta(\tilde t_i\mid x_i)+\log(t_i-t_{i-1}),
> $$
> so the gradient
> $$
> \nabla_\theta \mathrm{FOB}(t;\theta) \approx \sum_{i=1}^n  \nabla_\theta \log p_\theta(\tilde t_i\mid x_i)
> $$
> is driven by local density terms, which do not vanish.
>
> An empirical experiment shows that for $n=4,8,12$, the
> optimized FOB bound consistently tracks the full-order probability estimated by MC, especially for small n's.
>
> | n | MC estimate | FOB |
> |---|---:|---:|
> | 4  | 0.937 | 0.880 |
> | 8  | 0.673 | 0.537 |
> | 12 | 0.445 | 0.279 |
>
> To check the long-list performance more directly, we ran an additional experiment at n=256/512 using Kendall's
> $\tau$ as the evaluation metric. We obtain:
>
> | Method | n=256 |  n=512 |
> |---|------:|-------:|
> | RankNet | 86.06 |  87.29 |
> | ListMLE | 85.39 |  86.72 |
> | DiffSort | 67.80 |  63.75 |
> | FOB | 96.77 |  96.06 |
>
> This provides direct evidence that FOB still yields a meaningful learning signal for long lists.

---

> > ### Author Rebuttal · Reviewer_ewEE · 2026-04-02
> >
> > I think the writing should be improved end-to-end to make this clear and widely accessible.

---

> > > ### Author Response · Authors · 2026-04-04
> > >
> > > Thank you for taking the time to read our rebuttal and for clarifying that your remaining concern is primarily about the paper’s writing and accessibility.
> > >
> > > We appreciate this point, and we agree that the presentation should be improved substantially to make the paper easier to follow for a broader audience. In particular, as noted in our rebuttal, we will revise Section 3 carefully, especially the problem-definition part around lines 154–157, to make the motivation and setup explicit from the outset. We will also incorporate the clarifications from the rebuttal directly into the main text, rather than leaving them implicit.
> > >
> > > Regarding several of the writing-related concerns you raised (for example, specialized terminology and what the model outputs are), we would like to emphasize that these concepts are already explicitly defined in the original manuscript: the learning output is specified in Sec. 4.1, gradient-level non-separability is defined in Definition 3.1, the latent score model is introduced in Sec. 4.1, and bilevel optimization is introduced at the beginning of Sec. 5. That said, your feedback suggests that these definitions may not be sufficiently prominent or easy to locate in the current presentation. We take this seriously, and in the revision we will make these key definitions more visible and accessible, for example through stronger signposting, clearer formatting, and additional intuitive explanation around the formal statements.
> > >
> > > We also wanted to note one small point of clarification: in your review, you mention the term "intractable integrals." We checked the current manuscript and could not find this exact wording. It is possible that our presentation gave that impression, and if so, that again indicates we should explain the method more clearly and concretely in the revision.
> > >
> > > For the other concerns you raised earlier, such as bilevel optimization efficiency, the tightness of FOB, and the gradient signal for longer lists, we tried to address them in detail in the rebuttal with additional runtime, sensitivity, and long-list experiments. We hope those clarifications were helpful.
> > >
> > > In any case, we appreciate your candid feedback. We will take it seriously in the revision and work to improve the paper’s clarity end-to-end.

---

### Decision · Program_Chairs · 2026-04-30

**Decision:**

Accept (regular)

**Comment:**

The reviews are somewhat mixed, with two weak rejects and two positive recommendations (one weak accept and one accept). The rebuttal addresses many of the technical concerns and leads to increased confidence for the positive reviewers.

The main remaining concern is the clarity and presentation of the paper, particularly highlighted by one reviewer. However, this issue appears to be largely about exposition rather than fundamental technical flaws.

Given the situation, I recommend weak accept. If accepted, I strongly encourage the authors to significantly improve the presentation in the final version.